# Marriage of black phosphorus and Cu$^{2+}$ as effective photothermal agents for PET-guided combination cancer therapy

Kuan Hu[1], Lin Xie[1], Yiding Zhang[1], Masayuki Hanyu[1], Zhimin Yang[1,2], Kotaro Nagatsu[1], Hisashi Suzuki[1], Jiang Ouyang[3], Xiaoyuan Ji[3], Junjie Wei[2], Hao Xu[2], Omid C. Farokhzad[3], Steven H. Liang [4✉], Lu Wang [2,4✉], Wei Tao [3✉] & Ming-Rong Zhang[1✉]

The use of photothermal agents (PTAs) in cancer photothermal therapy (PTT) has shown promising results in clinical studies. The rapid degradation of PTAs may address safety concerns but usually limits the photothermal stability required for efficacious treatment. Conversely, PTAs with high photothermal stability usually degrade slowly. The solutions that address the balance between the high photothermal stability and rapid degradation of PTAs are rare. Here, we report that the inherent Cu$^{2+}$-capturing ability of black phosphorus (BP) can accelerate the degradation of BP, while also enhancing photothermal stability. The incorporation of Cu$^{2+}$ into BP@Cu nanostructures further enables chemodynamic therapy (CDT)-enhanced PTT. Moreover, by employing $^{64}$Cu$^{2+}$, positron emission tomography (PET) imaging can be achieved for in vivo real-time and quantitative tracking. Therefore, our study not only introduces an "ideal" PTA that bypasses the limitations of PTAs, but also provides the proof-of-concept application of BP-based materials in PET-guided, CDT-enhanced combination cancer therapy.

---

[1] Department of Advanced Nuclear Medicine Sciences, National Institute of Radiological Sciences, National Institutes for Quantum and Radiological Science and Technology, Chiba 2638555, Japan. [2] Department of Nuclear Medicine, PET/CT-MRI Center, The First Affiliated Hospital of Jinan University, Guangzhou 510630, China. [3] Center for Nanomedicine and Department of Anesthesiology, Brigham and Women's Hospital, Harvard Medical School, Boston 02115 MA, USA. [4] Division of Nuclear Medicine and Molecular Imaging, Massachusetts General Hospital, Harvard Medical School, Boston 02114 MA, USA. ✉email: liang.steven@mgh.harvard.edu; l_wang1009@foxmail.com; wtao@bwh.harvard.edu; zhang.ming-rong@qst.go.jp

As a minimally invasive and highly efficient therapeutic modality, cancer photothermal therapy (PTT) that uses photothermal agents (PTAs) to generate local hyperpyrexia for thermal elimination of tumors has yielded great success in both preclinical and clinical trials[1–5]. In a recent clinical study, PTT-induced ablation of tumors has demonstrated success in 94% of patients without deleterious changes in organ function, noticeable side effects, or severe complications[6]. With these promising results, as well as the clinical approval of iron oxide nanoparticles (NanoTherm, Magforce) for PTT in Europe, this therapeutic modality is expected to have not just a significant, but a revolutionary, clinical impact. However, it remains challenging to develop PTAs with both excellent photothermal effects (i.e., high photothermal conversion efficacy and/or photothermal stability for effective therapeutic outcomes) and rapid degradability (i.e., fast degradation for addressing safety concerns)[7,8].

In general, one type of PTAs commonly used in preclinical studies (e.g., inorganic nanomaterials) are based on nanoagents with relatively high photothermal stability under irradiation[9–13]. However, these PTAs usually degrade slowly or with difficulty, creating potential excretion problems and biosafety concerns[14–16]. Another commonly used PTA with rapid degradability (e.g., clinically approved indocyanine green) causes fewer biocompatibility and safety concerns. Nevertheless, rapid degradation compromises the photothermal stability which is necessary for superior therapeutic efficacy. Loss of photothermal function after only a few seconds or repeated irradiation is frequent[17–20]. Besides, laser irradiation can further exacerbate the degradation of PTAs. Though increasing the dosages of these PTAs may achieve more robust PTT, it would be accompanied by clinical problems such as the severe liver and kidney burden. Therefore, the conflict between high photothermal stability and rapid degradation of conventional PTAs represents a substantial impediment to fulfilling the clinical promise of PTT[21].

As a biocompatible and biodegradable two-dimensional (2D) nanomaterial, black phosphorus (BP) has been widely incorporated into useful PTAs with different nanostructures[22–24]. BP can also be easily oxidized and degraded to nontoxic phosphonates and phosphate, improving the safety of PTT[25–27]. Thus, BP is one of the PTAs with rapid degradability but unsatisfactory photothermal stability. Although several chemical modifications have been adopted to improve the stability and photothermal performance of BP[28–31], such gains come at the cost of prolonged degradation. Considering that: (i) BP is an inherent nanocaptor for Cu ions[32] (ii) Cu-based materials are also excellent PTA candidates[33,34], and (iii) $Cu^{2+}$ can accelerate the degradation of BP via redox reactions, we postulate that the synergy of BP and $Cu^{2+}$ may enhance the photothermal performance of BP@Cu nanostructures, while accelerating degradation.

Herein we report the development of a superior PTA based on BP nanosheets (BPNS), which tightly captures $Cu^{2+}$ to sidestep the aforementioned limitations common to PTAs in general. Specifically, the BPNS robustly captures $Cu^{2+}$ via coordination and electrostatic attraction. The $Cu^{2+}$ shell not only enhances the photothermal performance/stability of the BP-based PTAs, but also reacts with BPNS to accelerate degradation under irradiation. Besides, the attached Cu ions (Fenton-like reaction catalyst) can also react with hydrogen peroxide ($H_2O_2$) in the tumor microenvironment (TME) to generate cytotoxic hydroxyl radical (·OH), one of the most active reactive oxygen species (ROS)[35–38], enabling chemodynamic therapy (CDT)-concomitant PTT for enhanced therapeutic outcomes. Taking advantage of $^{64}Cu^{2+}$-PET functionality[39–43], we further demonstrate that this PTA can be used as a robust, versatile, and noninvasive imaging tool to track pharmacokinetics and monitor the therapeutic outcomes. We used a RGD-conjugated polyethylene glycol (PEG) to coat the developed BP@Cu nanostructure thus increasing the tumor tissue-specific accumulation and internalization for a better proof-of-principle application (Fig. 1). Taken together, we developed an intelligent strategy to resolve the previous impasse between photothermal stability and rapid degradation of PTAs. It is also notable that, although BP-based materials have been exploited extensively in various biomedical fields such as drug delivery[44–46], phototherapy and bioimaging[28,47–49], tissue engineering[50], and biosensing[51], this is the first report of the application of BP-based materials for PET-guided, CDT-enhanced combination cancer therapy.

## Results

### Synthesis and characterization of BP@Cu nanostructures.

Ultrasonication-assisted liquid exfoliation of bulk BP yields BPNS[52,53]. The transmission electron microscopy (TEM) image shows that the exfoliated BP possesses a sheet-like morphology and a lateral size of 60–120 nm (Fig. 2a and Supplementary Fig. 1a). The high-resolution (HR) TEM image reveals the characteristic spacing of 0.34 nm of the (021) lattice plane of the BP crystals[54] (Supplementary Fig. 1b). The atomic force microscopy (AFM) image reveals that the BPNS shows a statistically average lateral size of 80 nm (Fig. 2b and Supplementary Fig. 2) and an average thickness of $6 \pm 0.6$ nm, suggesting a stack of $10 \pm 1$ quintuple layers of BP[53] (Fig. 2c). The $Cu^{2+}$ capture experiment was performed based on methods described previously[32]. Elemental mapping revealed that the $Cu^{2+}$ was uniformly distributed on the BPNS (Fig. 2d, e, and Supplementary Fig. 3). This result was further confirmed by energy-dispersive X-ray spectroscopy (Supplementary Fig. 4), which shows intense signals from Cu. The TEM image of BP@Cu revealed that many holes appeared on the surface of BPNS (Supplementary Fig. 5a), leading to the blur of the crystal lattice of BPNS in the HR-TEM image (Supplementary Fig. 5b). This phenomenon may be attributed to the oxidation effect of $Cu^{2+}$.

We further modified BPNS with PEG to enhance stability and biocompatibility. Moreover, to confer the tumor-targeting ability to BPNS, a cyclic peptide c(RGDyC) that selectively binds to $\alpha_v\beta_3$ was conjugated to the PEG molecules[55]. Integrin $\alpha_v\beta_3$ is a subtype of the integrin family that is upregulated in both angiogenic endothelial cells and tumor cells, making it a particularly attractive therapeutic target[56]. The cyclic peptide c(RGDyC) was conjugated to the PEG molecules via the thiol-maleimide conjugation reaction[57]. Finally, a typical core-shell structured[58] nanosheet was prepared, as illustrated in Fig. 2f. The AFM image of BP@Cu@PEG-RGD showed that the BP@Cu was enveloped in a thin layer of PEG-RGD polymer, as the margin of the nanosheets was lighter in color than the core (Fig. 2g and Supplementary Fig. 6). The average size of BP@Cu@PEG-RGD nanosheets is approximately 150 nm, slightly larger than BPNS. The sectional plot shows that the thickness of the BP@Cu@PEG-RGD is between 10 and 20 nm (Fig. 2h), with the increase in thickness caused by the polymer coating. The TEM image of BP@Cu@PEG-RGD was shown in Supplementary Fig. 5c. No serious surface degradation was observed compared with BP@Cu (Supplementary Fig. 5d). The average hydrodynamic size of the BP@Cu@PEG-RGD nanosheets is 133.6 nm determined by dynamic light scattering (DLS) (Fig. 2i), which is within a range enabling efficient uptake by tumors due to the enhanced permeability and retention (EPR) effect.

### Synergistic photothermal effects between Cu ions and BPNS.

We synthesized a series of BP@Cu hybrid structures with different BPNS: $Cu^{2+}$ ratios, designated as BP@Cu$_x$ (the subscript number means "x" mmol $Cu^{2+}$ in one gram of BPNS; the same below unless otherwise specified). After 24 h incubation, the color

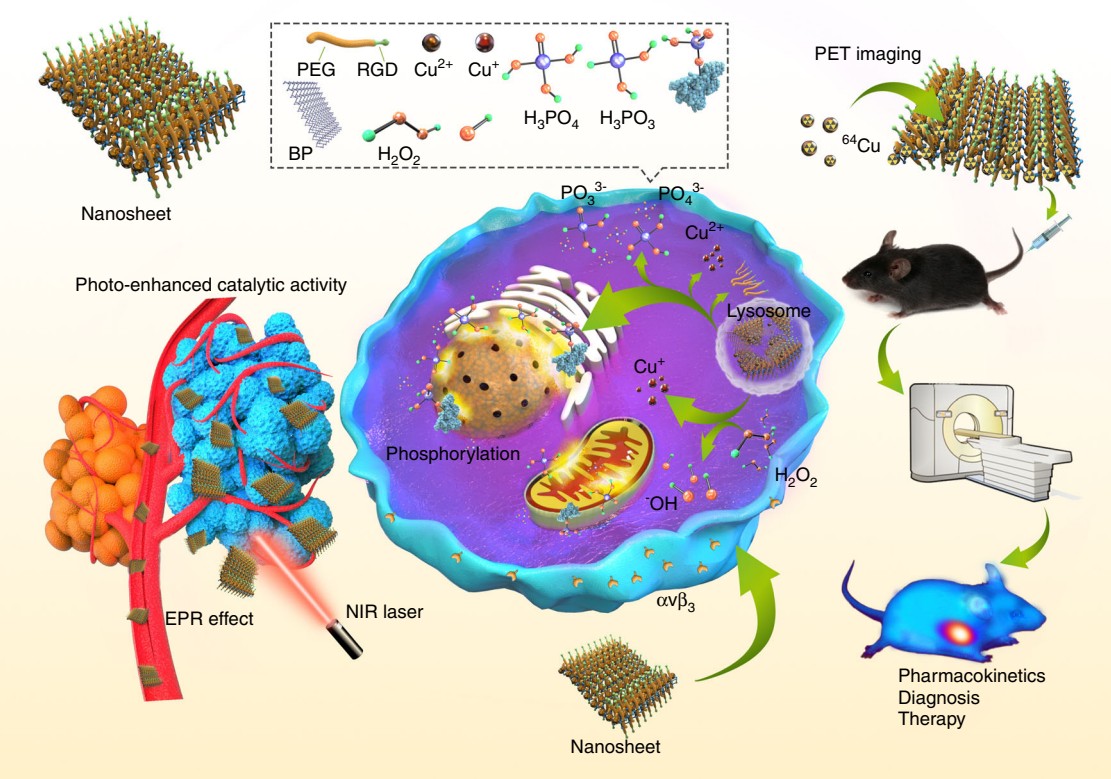

**Fig. 1 A multifunctional theranostic platform based on BP@Cu hybrid nanomaterial.** Inset box: illustration of the composition and construction of the BPNS-based nanomaterials. The biological responses elicited by the synergistic therapeutic effects of BPNS and Cu ions are illustrated. The BPNS is coupled with $^{64}$Cu radioisotope for PET imaging-guided pharmacokinetics and therapeutic efficacy studies. Abbreviations: EPR enhanced permeability and retention, NIR near-infrared, PET positron emission tomography.

of all the BPNS solutions turned from purple to lighter color, which depends on the concentration of $Cu^{2+}$. The BP@Cu$_{0.05}$ solution turned slightly lighter than that of BPNS, however, the BP@Cu$_{0.8}$ solution turned completely colorless, indicating serious degradation of the BPNS (Supplementary Fig. 7). An ultraviolet-visible-near infrared (UV-Vis-NIR) spectrometer was used to measure the absorbance of the samples. The coordination of $Cu^{2+}$ on the BPNS shows an obvious enhancement of the absorbance intensity of BP@Cu$_x$ compared with bared BPNS (Fig. 3a and Supplementary Fig. 8a), which is partially caused by the strong absorption of CuSO$_4$ solution at ~800 nm (Supplementary Fig. 8b). After 24 h incubation in water, all BP@Cu$_x$ samples revealed significantly lower absorption than bared BPNS at 300–1100 nm (Supplementary Fig. 8c), implying that the $Cu^{2+}$ facilitates the degradation of BPNS. Then we used AFM to examine the morphology of BP@Cu. As expected, the BPNS became thin; the average height of BPNS in BP@Cu$_{0.8}$ was ~0.5 nm after 24 h incubation, corresponding to a single layer of BP (Supplementary Fig. 9). To assess the degradation of BPNS after the addition of $Cu^{2+}$, the phosphate concentration in the supernatant was monitored[25]. The phosphate concentration in bare BPNS rose slowly; in contrast, the phosphate concentration was quickly elevated in the presence of $Cu^{2+}$, and the rate of increase was positively correlated with $Cu^{2+}$ concentration. Moreover, the oxidation of BPNS can be significantly facilitated by NIR laser irradiation (Fig. 3b).

Confocal Raman spectroscopy was performed to elucidate the influence of $Cu^{2+}$ modification. Pure BPNS creates three characteristic Raman peaks, the $A^1_g$, $B_{2g}$, and $A^2_g$ at around 360, 436, and 463 cm$^{-1}$, respectively[53] (Fig. 3c). For BP@Cu$_{0.2}$, the normalized peak intensity of all the three peaks shows decreased

compared with that of pure BPNS (the intensity of silicon transverse optical phonon of the substrate to be 1), indicating partial oxidation of the BPNS[59]. Moreover, a redshift of the Raman peaks was observed, which is attributed to the formation of either P-O bonds or electrostatic interactions between P atom and $Cu^{2+}$ ions. After modification by PEG-RGD, the Raman intensity recovered to the level of pure BPNS while exhibiting a further red-shift of the peaks (Fig. 3c). X-ray photoelectron spectroscopy (XPS) was employed to further study the interaction details between $Cu^{2+}$ and BPNS. In the high-resolution P $2p$ spectrum of BPNS, the intense peak centered at 130 eV is attributed to the $2p_{3/2}$ and $2p_{1/2}$ of P–P bonds, which can be deconvoluted to 130.4 and 131.5 eV, respectively[60] (Fig. 3d and Supplementary Fig. 10). Another broad peak appeared at 132.5–137.2 eV, which is caused by the unavoidable oxidation of BPNS to P$_x$O$_y$ species. For BP@Cu$_{0.2}$, there was only an intense peak centered around 135 eV, indicating the complete degradation of BPNS. Besides, the Cu $2p$ XPS spectrum was analyzed. The BPNS shows no peak in this area; however, the BP@Cu$_{0.2}$ shows two peaks at 933 and 935 eV corresponding to the P $2p_{2/3}$ of $Cu^+$ and $Cu^{2+}$, respectively (Fig. 3e). The Cu $2p_{1/2}$ peaks of $Cu^+$ and $Cu^{2+}$ appear at 953 and 958 eV, respectively[32]. The ratio of $Cu^{2+}$ and $Cu^+$ is about 1:1.4 (Supplementary Fig. 11), indicating most of the $Cu^{2+}$ was reduced to $Cu^+$. The reduction of $Cu^{2+}$ was further confirmed by electron paramagnetic resonance (EPR) study. The CuSO$_4$ showed an intense paramagnetic signal but the BPNS shows no signal (Supplementary Fig. 12a). When measure the EPR spectra of BP@Cu$^{2+}$ at different time points in atmosphere, apparent decreases of the signal intensities were observed (Supplemental Fig. 12b), indicating the reduction of $Cu^{2+}$ to non-paramagnetic $Cu^+$ charge state.

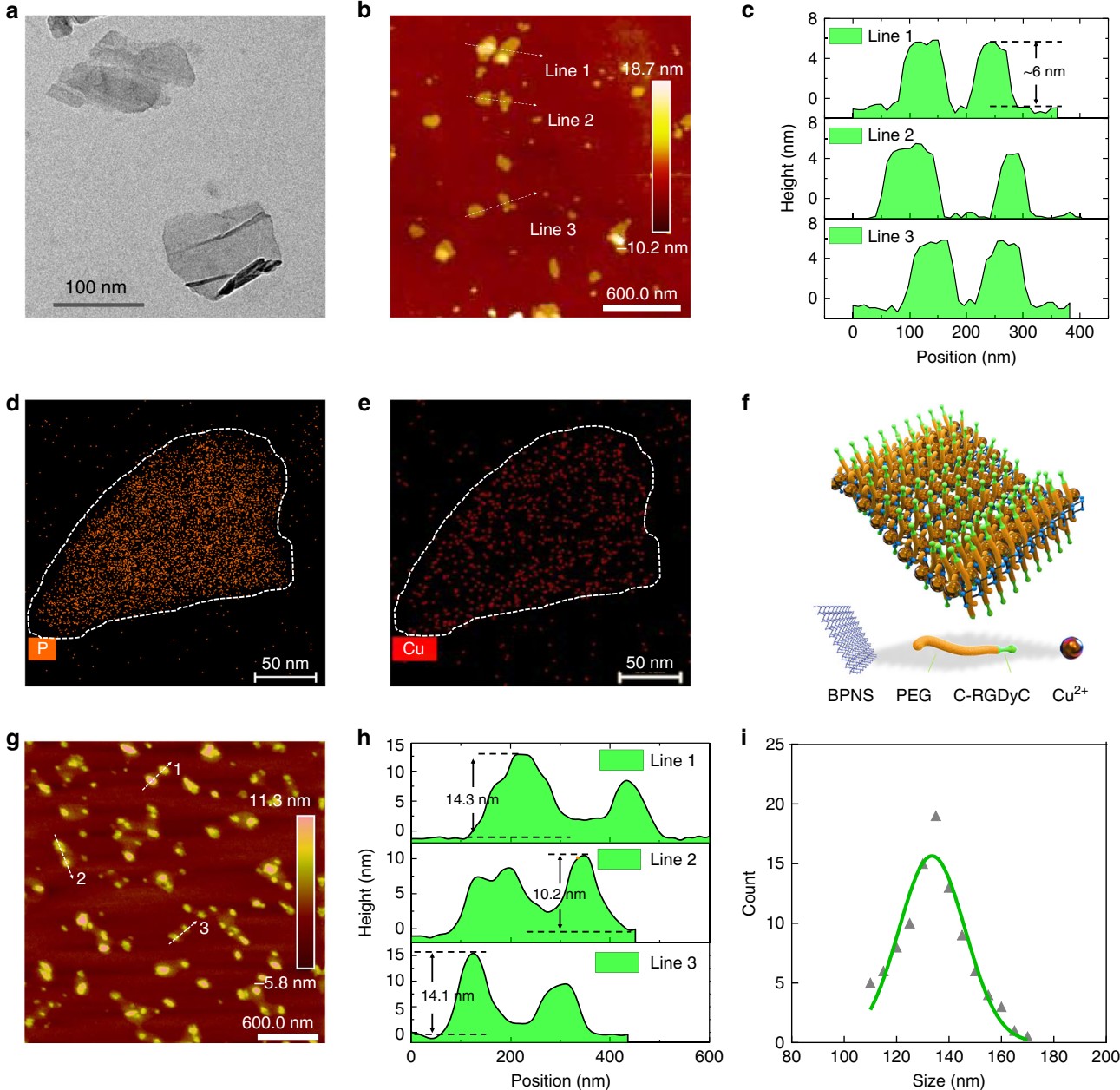

**Fig. 2 Characterization of BPNS-based nanomaterials. a** TEM image of BPNS. Similar TEM images of BPNS were obtained for more than three times experiments. **b** AFM image of BPNS. **c** Height profiles of BPNS along the white lines in (**b**). **d**, **e** STEM elemental mapping of P and Cu for BP@Cu hybrid nanosheets, respectively. Similar STEM mapping images were observed for more than three independent experiments. **f** Schematic diagram of the composition and structure of BP@Cu@PEG-RGD. **g** AFM image of BP@Cu@PEG-RGD. **h** Height profiles of BP@Cu@PEG-RGD along the white lines in (**g**). **i** DLS measurements of the hydrodynamic diameter of BP@Cu@PEG-RGD (0.1 mg/mL of BPNS) in PBS. Abbreviations: AFM atomic force microscopy, TEM transmission electron microscopy, STEM scanning TEM, DLS dynamic light scattering, PBS phosphate-buffered saline.

We further investigated the photothermal conversion efficiency (PTCE) of BPNS after loading of $Cu^{2+}$ ions. After ten min irradiation by a NIR laser. the temperature of BPNS, BP@Cu$_{0.2}$, and BP@Cu$_{0.2}$@PEG-RGD increased by 16 °C, 28 °C, and 31 °C, respectively (Fig. 3f), resulting in the corresponding temperature increments per minute for them are 1.4, 2.4, and 2.7 °C min$^{-1}$ (Supplementary Fig. 13).The IR thermal images of the samples confirmed these results (Fig. 3g). The PTCEs were approximately 28.7, 33.6, and 35.4% for BPNS, BP@Cu$_{0.2}$, and BP@Cu$_{0.2}$@PEG-RGD, respectively[61]. Notably, the superior PTCE of BP@Cu$_{0.2}$@-PEG-RGD compared with BP@Cu$_{0.2}$ can be ascribed to the partial degradation of BPNS in the BP@Cu$_{0.2}$. The loading mass of $Cu^{2+}$ ions on BPNS also affects the PTCE. As shown in Fig. 3h, the

temperature increment of BP@Cu$_x$@PEG-RGD is positively correlated to the $Cu^{2+}$ mass loading on BPNS, and more $Cu^{2+}$ results in better photothermal conversion. These results suggest the possible photothermal synergistic effects between BPNS and $Cu^{2+}$. First, the $Cu^{2+}$ shell on BPNS helps to transfer the heat from the BPNS to the surrounding environment. Second, the Cu ions may aggregate on the surface of the BPNS to become additional photothermal conversion centers. Third, the redox reaction between BPNS and $Cu^{2+}$ may release a huge amount of heat.

**Synergistic effects of $Cu^{2+}$ and BPNS on killing cancer cells.** The influence of $Cu^{2+}$ on the cellular response to BPNS was

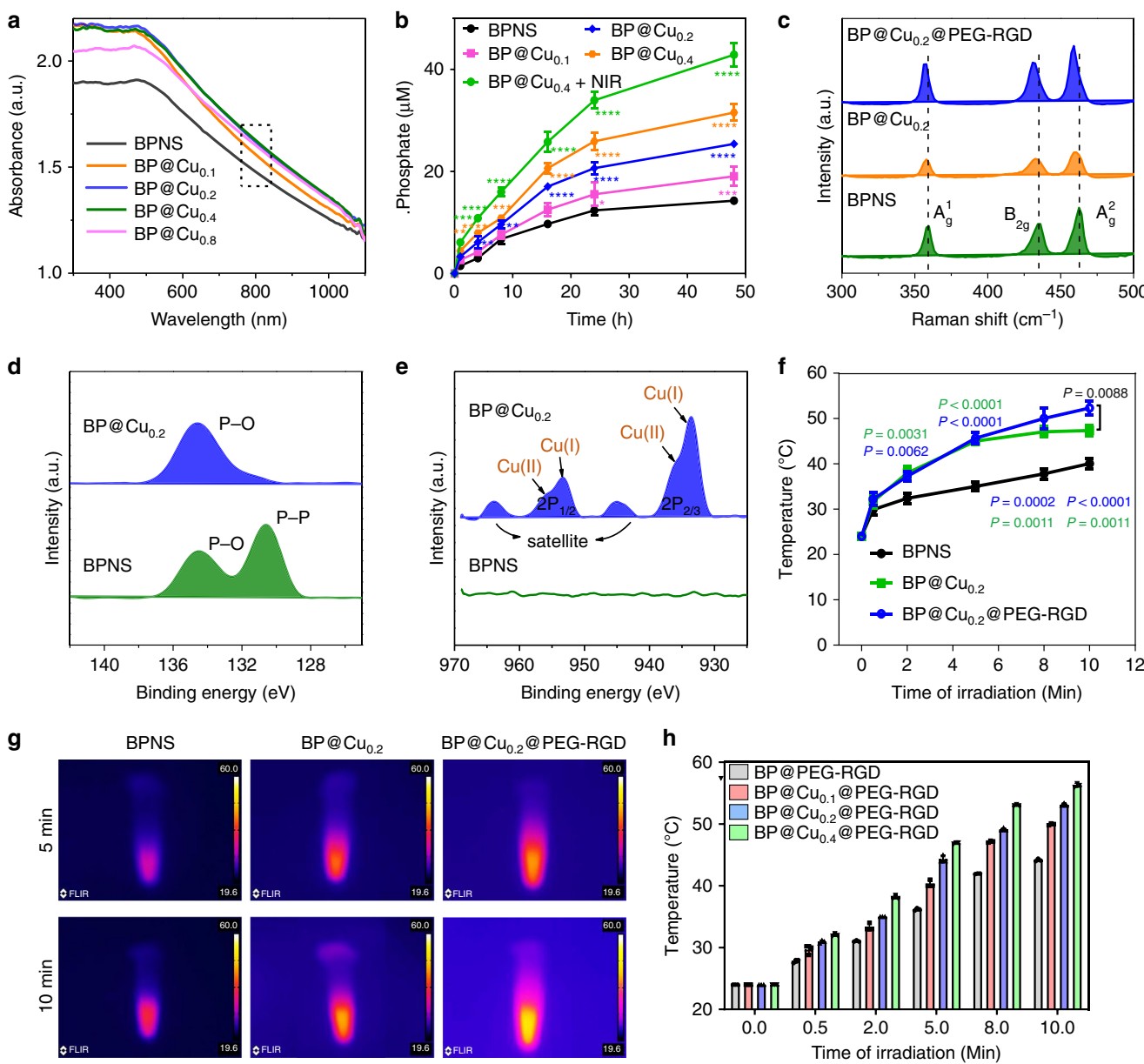

**Fig. 3 Influence of Cu$^{2+}$ ions on the stability and photothermal effects of BPNS. a** UV–vis absorbance spectra of BPNS, BP@Cu$_{0.1}$, and BP@Cu$_{0.2}$ and BP@Cu$_{0.4}$ with the same amount of BPNS (100 ppm). **b** Measurement of phosphate anions by phosphate assay kit. Degradation of BPNSs and BP@Cu$_x$ with the same amounts of BPNS (50 ppm) after storage in water for different periods, producing increasing concentrations of phosphate anions in the supernatant. The data show mean ± s.d. $n = 3$. *$P < 0.05$, **$P < 0.01$, ***$P < 0.001$, ****$P < 0.0001$, analyzed by one-way ANOVA, followed by Tukey's multiple comparisons post-test. **c** Raman scattering spectra of BPNS, BP@Cu$_{0.2}$, and BP@Cu$_{0.2}$@PEG-RGD. **d** High-resolution XPS spectra showing the binding energies of P$_{2p}$ of BPNS and BP@Cu$_{0.2}$. **e** High-resolution XPS spectra showing the binding energy of Cu$_{2P}$ of BPNS and BP@Cu$_{0.2}$. **f** Photothermal heating curves recording the temperature variations of BPNS, BP@Cu$_{0.2}$, and BP@Cu$_{0.2}$@PEG-RGD with the same amount of BPNS (20 ppm) dispersed in PBS. The data show mean ± s.d., $n = 3$. $P$ values marked in green and blue colors indicate statistically significant differences compared with the BPNS group. $P$ values marked in black value indicates statistically significant differences between BP@Cu$_{0.2}$ and BP@Cu$_{0.2}$@PEG-RGD. Analyzed by one-way ANOVA, followed by Tukey's multiple comparisons post-test. **g** Infrared thermographic maps of BPNS, BP@Cu$_{0.2}$, and BP@Cu$_{0.2}$@PEG-RGD with the same amounts of BPNS (20 ppm) in tubes after irradiation for 5 and 10 min under the 808 nm laser (1 W cm$^{-2}$). **h** The temperature increments of different BP@Cu$_x$@PEG-RGD samples under NIR laser irradiation by an 808 nm laser for different periods. The data represents the mean ± s.d. of three independent experiments.

evaluated. The acute degradation of BPNS in cancer cells can yield high levels of phosphate anions and immediately lead to the death of cancer cells[48,62]. We conceived that the therapeutic effects of BPNS could be precisely controlled by the addition of Cu$^{2+}$ ions and NIR irradiation, as they co-modulate the degradation of BPNS. To test this hypothesis, mouse melanoma B16F10 cells were exposed to different BP species, and the cell

viability was measured. Pure BPNS showed very a slight inhibitory effect on the proliferation of B16F10. In contrast, the proliferation activity of B16F10 treated with 100 μg mL$^{-1}$ of BP@Cu$_{0.2}$@PEG-RGD or BP@Cu$_{0.2}$ remained ~45% of the BPNS treated cells. Moreover, the cells treated with 100 μg mL$^{-1}$ of BP@Cu$_{0.2}$@PEG-RGD and exposed to 808 nm NIR irradiation demonstrated the strongest inhibition, retaining only ~10% of

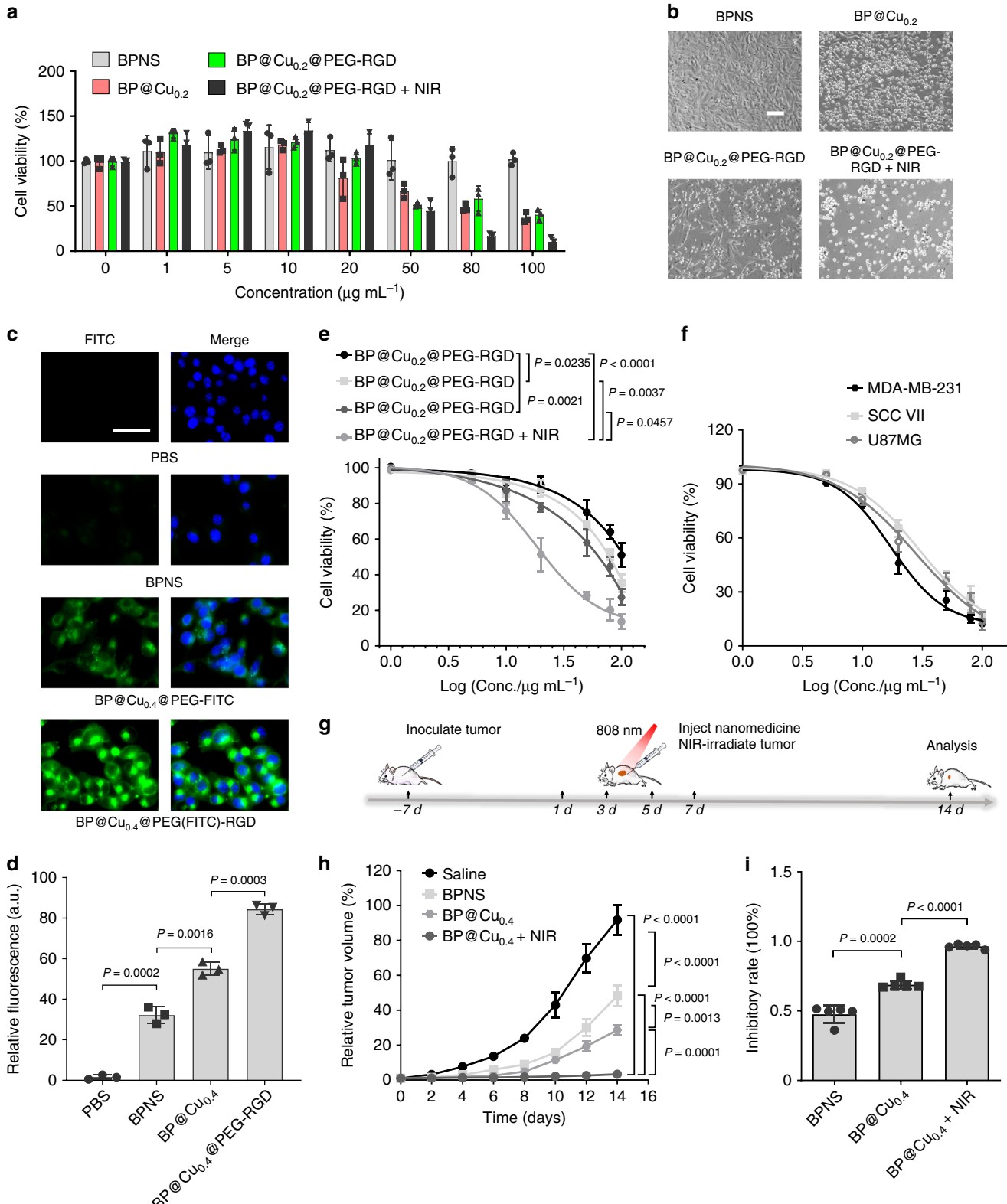

proliferation activity of the BPNS treated cells (Fig. 4a, b). The cancer cell killing effect of the nanocomplex prerequisites the effective cellular uptake of the nanocomplex, thus, we conducted confocal fluorescence imaging in live B16F10 cells to study the cellular uptake of the nanocomplexes. Live-cell fluorescent images from 6-h incubation showed that both the nanocomplexes enter

the cancer cells successfully (Fig. 4c, Supplementary Figs. 14, 15), and BP@Cu$_{0.4}$@PEG(FITC)-RGD is more likely to be uptaken by B16F10 cells (Fig. 4d). The relationship between the anti-proliferation effect and the amount of Cu$^{2+}$ loaded on BPNS was then investigated. The viability of B16F10 cells was Cu$^{2+}$ mass-dependent, and the anti-proliferation effects ranked as follows:

**Fig. 4 Synergistic effects of BPNS and Cu$^{2+}$ ions on killing cancer cells. a** Cell viabilities measured by MTT assay. B16F10 cells were subjected to various concentrations of BPNS, BP@Cu$_{0.2}$, BP@Cu$_{0.2}$@PEG-RGD, and BP@Cu$_{0.2}$@PEG-RGD + NIR ($n = 3$). **b** Live-cell differential interface contrast imaging of B16F10 cells after incubation with BPNS, BP@Cu$_{0.2}$, BP@Cu$_{0.2}$@PEG-RGD, or BP@Cu$_{0.2}$@PEG-RGD + NIR with the same amount of BPNS (100 ppm) for 48 h, respectively. Scale bars, 100 μm for all panels. **c** Confocal fluorescence images of live B16F10 cells incubating with PBS, BPNS (100 ppm), BP@Cu$_{0.4}$@PEG-FITC (100 ppm BPNS), or BP@Cu$_{0.4}$@PEG(-FITC)-RGD (100 ppm BPNS) for 6 h. Scale bar, 50 μm for all panels. Similar confocal fluorescence images were obtained for three independent experiments. **d** Relative FITC fluorescence in the cells. Mean ± s.d., $n = 3$. Unpaired two-tailed student's $t$-test. **e** Comparison of the 48 h cytotoxicity of BP@Cu$_{0.1}$, BP@Cu$_{0.2}$, BP@Cu$_{0.4}$, and BP@Cu$_{0.4}$ plus 2 min of NIR laser irradiation against B16F10 cells ($n = 3$, analyzed by one-way ANOVA, followed by Tukey's multiple comparisons post-test). **f** Relative cell viabilities of MDA-MB-231, SCC VII, and U87MG cancer cells after incubation with different concentrations of BP@Cu$_{0.4}$@PEG-RGD. NIR laser irradiation was applied for 2 min after 4 h of incubation ($n = 3$). **g** Treatment schedule for in vivo cancer therapy by intratumoral injection of nanomaterials. **h** Relative tumor growth curves of the tumor receiving different treatments. Mice were intratumorally injected with 100 μL saline, BPNS, or BP@Cu$_{0.4}$. One group of mice injected with BP@Cu$_{0.4}$ received NIR laser irradiation at a density of 1 W cm$^{-2}$ for 2 min at 4 h postinjection. The data represent mean ± s.d. ($n = 5$), analyzed by two-way ANOVA, followed by Tukey's multiple comparisons post-test. (**i**) Inhibitory rates of B16F10 tumors at day 14 post-treatment. Mean ± s.d., $n = 5$. Analyzed by the unpaired two-tailed student's $t$-test. All error bars in this figure indicate standard deviation.

BP@Cu$_{0.4}$ > BP@Cu$_{0.2}$ > BP@Cu$_{0.1}$ (Fig. 4e). Moreover, the 808 nm NIR irradiation impressively enhanced the killing effects of BP@Cu$_{0.4}$@PEG-RGD, as the IC$_{50}$ value of BP@Cu$_{0.4}$@PEG-RGD + NIR (~31.6 μg mL$^{-1}$) decreased to about one-third of that of BP@Cu$_{0.4}$@PEG-RGD (~100 μg mL$^{-1}$) (Fig. 4e). Then the cytotoxicity of BP@Cu$_{0.4}$@PEG-RGD was tested in three other kinds of tumor cells (MDA-MB-231, SCC VII, and U87MG). All cells displayed dose-dependent cell susceptibility to BP@Cu$_{0.4}$@PEG-RGD + NIR irradiation treatment (Fig. 4f), and the IC$_{50}$ values ranged from 30 to 50 μg mL$^{-1}$ according to cell type. These results together demonstrate that Cu$^{2+}$ enhances the anti-proliferation effect of BPNS on cancer cells and that NIR irradiation can further strengthen the synergistic effects.

Encouraged by the in vitro results, we further evaluated the anti-tumor effects of BP@Cu in vivo using B16F10-bearing mice as models. The treatment plan was shown in Fig. 4g. During the treatment period, we observed few symptoms of toxic side effects in the experimental groups. The mice injected with BPNS showed moderate tumor growth inhibition on day 14 compared with mice injected with saline (Fig. 4h). The corresponding tumor inhibitory rate in this group is ~50% (Fig. 4i). In contrast, the mice injected with BP@Cu showed remarkable tumor growth inhibition, >70%. Significantly, the mice that were injected with BP@Cu and received NIR laser irradiation demonstrated the most robust effects, with a tumor inhibitory rate of >95% (Fig. 4i).

**Mechanism study of BP@Cu on killing cancer cells.** To understand the effects of Cu$^{2+}$ ions on the cellular responses of BPNS, we first studied apoptosis in cancer cells after treatment with BP@Cu species[48]. Annexin V-FITC/Propidium iodide (PI) double staining and fluorescence-activated cell sorting (FACS) analysis were used to characterize apoptosis. As seen in Fig. 5a, b, BPNS treatment elicited a higher apoptosis rate in B16F10 cells (17.8%, sum of Annexin V-FITC+/PI+ and Annexin V-FITC+/PI−) compared with PBS treatment (10.2%). The addition of Cu$^{2+}$ further increased apoptosis to 38.1% in BP@Cu-treated cells. Significantly, the cells treated with BP@Cu@PEG-RGD demonstrated the highest apoptosis rate (61.5%). To confirm this result, TMRE (tetramethylrhodamine, ethyl ester) was used to label active mitochondria and acts as an indicator for mitochondria trans-membrane potential (Δψm) changes of the cells, as the changes of Δψm can lead to the release of apoptogenic factors and loss of oxidative phosphorylation[63]. For cells treated with PBS, bright red fluorescence was emitted from the cell plasma. In contrast, the fluorescence intensity was sharply decreased in almost all cells treated with BP@Cu$_{0.4}$@PEG-RGD, suggesting the significant loss of Δψm of the cells, compared with the decrease in only a small portion of cells treated with BP@Cu0.4 (Supplementary Fig. 16). The cell apoptosis is executed through two major executioner

caspases, caspase-3 and caspase-7 (ref. [64]). Therefore, we examined the caspase-3/7 activation in B16F10 cells after treatment with BPNS nanomaterials. For cells treated with PBS or BPNS alone, few green fluorescence positive cells (caspase-3/7 activated cells) were observed. In contrast, the BP@Cu$_{0.4}$ or BP@Cu$_{0.4}$@PEG-RGD treatments dramatically increased the population of green fluorescence positive cells (Fig. 5c, d, and Supplementary Fig. 17), indicating the strong effects on activation of caspase-3/7. The activation of caspase-3 can initiate apoptotic DNA fragmentation[65]. The cell apoptosis was also detected with the TUNEL (terminal deoxynucleotidyltransferase dUTP nick and labeling) assay. As seen in Supplementary Fig. 18, the highest Apo-BrdU positive cell portions were determined in BP@Cu$_{0.4}$@PEG-RGD-treated group, in good agreement with the results in Annexin V-FITC/PI staining and caspase-3/7 detection.

Since programmed cell death is connected to cell cycle arrest, we analyzed the cell cycle distribution of B16F10 cells treated with different BPNS reagents using PI staining. As shown in Fig. 5e, f, exposure of B16F10 to BPNS increased the percentage of G2/M phase cells (14.9% in G2/M) compared with the percentage of PBS-treated cells (8.1%). Moreover, BP@Cu$_{0.4}$ treatment caused a further larger G2/M phase arrest (28.7%). As expected, cells treated with BP@Cu$_{0.4}$@PEG-RGD exhibited the highest G2/M phase arrest (33.0% in G2/M), accompanied by a large decrease in the number of cells in the G0/G1 phase. The changes in cell cycle distribution were further confirmed by using DyeCycle dye staining in live cells (Supplementary Fig. 19).

Cu(I/II) is an attractive Fenton-like catalyst to generate highly cytotoxic ROS, such as the hydroxide radical ·OH, which induce cell apoptosis[37]. In our system, the Cu$^{2+}$ loaded on BPNS can be reduced to Cu$^+$, which may react with the local H$_2$O$_2$ via a Fenton-like reaction to generate toxic hydroxyl radicals (·OH). Besides, the Cu$^{2+}$ can accelerate GSH depletion, weakening the ROS scavenging ability of GSH in tumor cells (Supplementary Fig. 20). To confirm this hypothesis, we analyzed the total cellular oxidative stress and superoxide species after treatment with different BPNS species. As shown in Fig. 5g, h, BPNS treatment elevated the ROS level of the cancer cells (9.2%) compared with that of PBS-treated cells (2.9%), and BP@Cu$_{0.4}$ treatment further increased the ROS level (11.7%). Consistently, cells treated with BP@Cu$_{0.4}$@PEG-RGD generated the highest ROS levels (23.7%). The intracellular ROS generation was further confirmed by DCFH-DA staining (Supplementary Fig. 21a). The brightest fluorescence was observed in BP@Cu$_{0.4}$@PEG-RGD treated cells, indicating the highest ROS concentration in cells. The BP@Cu$_{0.4}$ treated cells showed brighter fluorescence than BPNS-treated cells, suggesting that the Cu$^{2+}$ plays a vital role in inducing the production of ROS in cells. Then Sytox green was used to stain the dead cells and inspect the intact cell membranes. As seen in

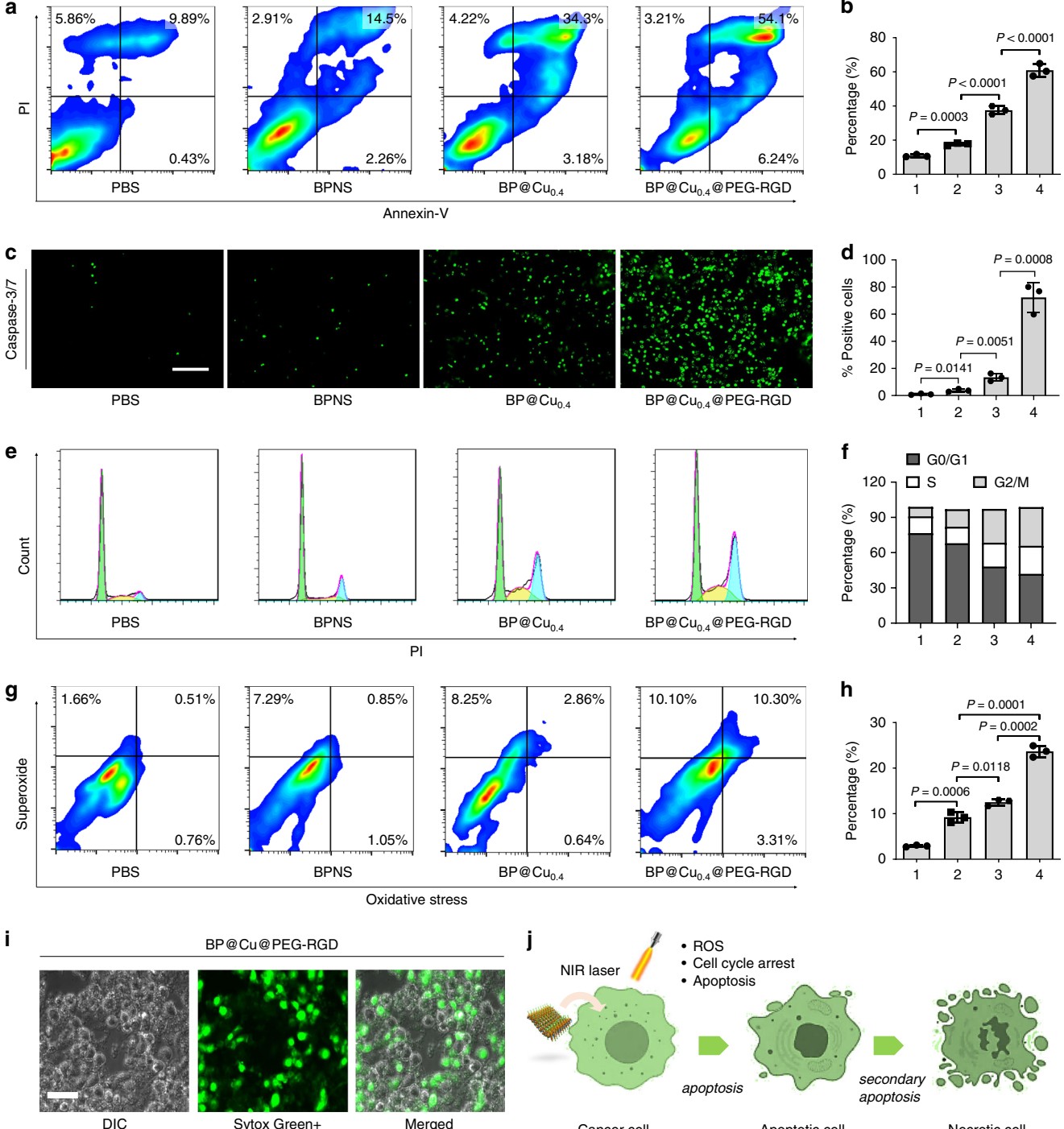

**Fig. 5 Cellular effects of BP@Cu$_{0.4}$@PEG-RGD in cancer cells.** B16F10 cells were treated with (1) PBS (control), (2) BPNS (100 ppm), (3) BP@Cu$_{0.4}$ (100 ppm of BPNS), or (4) BP@Cu$_{0.4}$@PEG-RGD (100 ppm of BPNS) for 24 h. NIR irradiation (808 nm, 1 W cm$^{-2}$, 2 min) was applied to all cells after incubation for 4 h. **a** Annexin-V and PI double-staining flow cytometry analysis of the apoptosis of B16F10 cells. The data was processed by the Flowjo program. **b** Quantitative analysis of apoptotic cells after different treatments. The number 1–4 indicates was assigned to treatment groups indicated before. The same below. The data represent Annexin-V (+) cells. mean ± s.d., $n = 3$, unpaired two-tailed student's $t$-test. **c** Activation of caspase-3/7 in B16F10 cells after treatment of PBS, BPNS, BP@Cu$_{0.4}$, or BP@Cu$_{0.4}$@PEG-RGD for 24 h. Scale bars, 100 μm for all panels. **d** Quantification of caspase-3/7 positive cells. The data represents mean ± s.d., $n = 3$ biologically independent experiments, unpaired two-tailed student's $t$-test. **e** Cell-cycle distribution of B16F10 cells stained with propidium iodide and analyzed by flow cytometry. The data was processed by the Flowjo program. **f** Summary of cell percentage in each cell cycle phase, (1) PBS, (2) BPNS, (3) BP@Cu$_{0.4}$, (4) BP@Cu$_{0.4}$@PEG-RGD. **g** Flow cytometry analysis of cellular ROS/superoxide levels in B16F10 cells after different treatments. **h** Percentage of cells showing increased ROS/superoxide level after treatments. mean ± s.d., $n = 3$, unpaired two-tailed student's $t$-test. **i** Sytox Green staining of B16F10 cells after treatment with BP@Cu$_{0.4}$@PEG-RGD + NIR irradiation. Similar Sytox Green staining images were obtained for three independent experiments. **j** Probably cellular response after treatment with BP@Cu$_{0.4}$@PEG-RGD.

Fig. 5i, most cells treated with BP@Cu$_{0.4}$@PEG-RGD had impaired membranes and displayed abundant apoptotic bodies. However, this phenomenon was much subtler in cells treated with PBS, BPNS, or BP@Cu0.4 (Supplementary Fig. 21b). In sum, Cu$^{2+}$ ions elicit dramatic changes in the cancer cell's biological activity, e.g., accelerating the degradation of BPNS and enhancing the photothermal effect of BPNS. Furthermore, the Cu$^{2+}$ released from the nanoparticles feeds the redox cycle in cancer cells and induces robust cytotoxic ROS (Fig. 5j).

**Pharmacokinetics of BP@Cu nanostructures**. The superior therapeutic effects and biocompatibility of BPNS make it an excellent nanomaterial for cancer therapy. However, the detailed in vivo drug metabolism and pharmacokinetics (DMPK) of BPNS-based nanodrugs have not been fully clarified. Although several studies have tried to elucidate the biodistribution of BPNS-based nanomaterials[28,45], optical-based imaging methods, such as photoacoustic imaging (PAI)[31], are limited in terms of in real-time, quantitative and full-body mapping of the targeted compounds. Among existing imaging techniques, PET imaging is a tool to noninvasively assess the pharmacological properties and pharmacokinetics of a variety of drug molecules, given its excellent quantitative capability, unlimited signal penetration, and high detection sensitivity[66,67]. In this context, it is conceivable to combine PET imaging functionality with BPNS for accurate quantitative tracking of in vivo behavior in real-time.

The labeling of BPNS with $^{64}$Cu is straightforward using the "chelator-free" method[68]. The entire labeling process was completed in 10 min and the labeling efficiency reached 99%. The molar activity of BP@$^{64}$Cu is established to be ~74 MBq μg$^{-1}$ of BPNS. The stability of BP@$^{64}$Cu complex was examined by using radio-thin layer chromatography (radio-TLC) in PBS and mouse serum. As shown in Fig. 6a and Supplementary Figs. 22–24, the BP@$^{64}$Cu was very stable in PBS as less than 5% degradation was observed up to 70 h. In mouse serum, the BP@$^{64}$Cu@PEG-RGD showed gradual degradation with the increase of incubation time. The half-life in serum is estimated to be 35 h. The uptake of BP@$^{64}$Cu nanomaterials by B16F10 cells was investigated. From 0 to 80 min, the BP@$^{64}$Cu@PEG-RGD gradually accumulated in cancer cells. However, the uptake of BP@$^{64}$Cu@PEG was remarkably decreased to approximately one fourth that of BP@$^{64}$Cu@PEG-RGD. Moreover, the co-incubation of c(RGDyC) peptide with BP@$^{64}$Cu@PEG-RGD also inhibited the uptake of the nanosheets (Fig. 6b). A competition binding assay revealed that the uptake of BP@$^{64}$Cu@-PEG-RGD can be blocked by free c(RGDyC) peptide or BP@Cu@PEG-RGD nanosheets in a concentration-dependent manner (Fig. 6c).

Then in vivo PET imaging was performed in mice inoculated with B16F10 melanoma tumors. After intravenous injection (i.v.) of BP@$^{64}$Cu@PEG-RGD, the whole tumor demonstrated uptake within 8 h, and the highest uptake was about 5.2% ID g$^{-1}$ at 16 h post-injection (Fig. 6d, g). The radiolabeled BP@$^{64}$Cu@PEG-RGD showed long-term retention in tumors, with ~4.8% ID g$^{-1}$ remaining at 40 h post-injection (Fig. 6g). Compared with BP@$^{64}$Cu@PEG-RGD, BP@$^{64}$Cu@PEG showed much lower specific uptake in the tumor according to the maximum intensity projection (MIP) PET images (Fig. 6e). Furthermore, the tumor accumulation of BP@$^{64}$Cu@PEG-RGD was significantly reduced by coadministration of excess c(RGDyC) peptides (Fig. 6f). In summary, these results confirm that the uptake of BP@$^{64}$Cu@-PEG-RGD (BP@Cu@PEG-RGD) is dominated by the c(RGDyC)-mediated active targeting effect, in accord with the in vitro results.

The detailed pharmacokinetics of BP@$^{64}$Cu@PEG-RGD was then analyzed by ex vivo biodistribution. The highest radioactivity appeared in the spleen, followed by the liver and lung

(Fig. 6h), in good consistency with the PET imaging. The uptake of BP@$^{64}$Cu@PEG-RGD in tumors continuously increased from 0 to 24 h post-injection (Fig. 6i), and the highest tumor uptake was 9.3% ID per gram tissue. In contrast, the nanotracer reached its highest accumulation in spleen, lung, and liver in the first 3 h and then was gradually excreted from these organs, reflected by a sharp decrease in intensity from 3 to 18 h post-injection (Fig. 6h). The excretion of the nanotracer by the liver, spleen, and lung might be ascribed to the fast oxidation of the BPNS in these oxygen-rich organs[69]. Notably, the %ID g$^{-1}$ in the blood persistently increased from 1 to 42 h following i.v. injection. This phenomenon can be explained by the reversible uptake of BP@Cu@PEG-RGD in the mononuclear phagocytes. Specifically, the BP@Cu@PEG-RGD is initially uptaken by mononuclear phagocytes because of the large size (~134 nm). With the oxidative degradation of BPNS, the BP@Cu@PEG-RGD would reduce its size (<35 nm) and eventually escape from the RES-rich organs re-enter the blood circulation. The kidney uptake could also be associated with possible renal excretion, as evidenced by considerable radioactivity in the urine and bladder. To preclude the radioactivity in the tumor from the free $^{64}$Cu ions, we performed PET imaging study and ex vivo biodistribution of free $^{64}$Cu ions in the mouse model (Supplementary Figs. 25 and 26). Both experiments revealed that the $^{64}$Cu$^{2+}$ differed significantly from BP@$^{64}$Cu@PEG-RGD on DMPK, confirming considerable stability of BP@$^{64}$Cu@PEG-RGD in vivo. Finally, the tumor-targeting ability of BP@$^{64}$Cu@PEG-RGD was further investigated in SCC VII tumor and MDA-MB-231 tumor models. As shown in Fig. 6j and Supplementary Fig. 27, the BP@$^{64}$Cu@PEG-RGD demonstrated specific uptake in both kinds of tumors, regardless of tumor size. These results collectively demonstrate that BPNS is an excellent nanomaterial for tumor diagnosis and therapy.

**In vivo combinatorial cancer therapy**. Encouraged by the excellent tumor-targeting ability of BP@$^{64}$Cu@PEG-RGD, we further studied the in vivo antitumor therapeutic efficacy of the nanomaterials by i.v. injection. The treatment plan was shown in Fig. 7a. To monitor the photothermal effects in situ, we employed an infrared thermal imaging camera to record thermographic maps and changes in tumor temperature (Fig. 7b and Supplementary Fig. 28). Under the NIR laser irradiation, the temperature of tumors in mice injected with BP@Cu$_{0.4}$@PEG-RGD nanomaterials rose rapidly by 30 °C within 5 min to a maximum of 56 °C, sufficient for thermal ablation. In contrast, the tumors injected with BP@Cu$_{0.4}$ or BP@Cu$_{0.4}$@PEG showed slighter temperature elevation (16 and 19 °C, respectively). This difference can be explained by variation in specific uptake of nanodrugs by tumors. As shown in the tumor growth curves in Fig. 7c–f, both BP@Cu$_{0.4}$ and BP@Cu$_{0.4}$@PEG treatments failed to effectively suppress tumor growth, attributable mainly to low uptake of nanodrugs. Effective tumor growth inhibition was achieved in mice treated with BP@Cu$_{0.4}$@PEG-RGD plus 808 nm NIR laser irradiation. In this group, the tumor inhibitory effect was consistent in all six mice, with relative tumor volume about 10% of that in the saline-treated group (Fig. 7g). The satisfactory antitumor efficacy of BP@Cu$_{0.4}$@PEG-RGD + PTT can be attributed to the high uptake of nanodrugs in the tumor, which induces robust intratumoral ROS and intense photothermal effects. The body weight of the mice increases steadily in all four groups without noticeable detrimental side effects (e.g., abnormality in body weight or death), suggesting satisfiable biocompatibility and little in vivo toxicity of BP@Cu nanomaterials (Fig. 7h).

Finally, therapeutic efficacy was confirmed by PET imaging, which showed that tumors in saline-treated mice (Fig. 7i) were

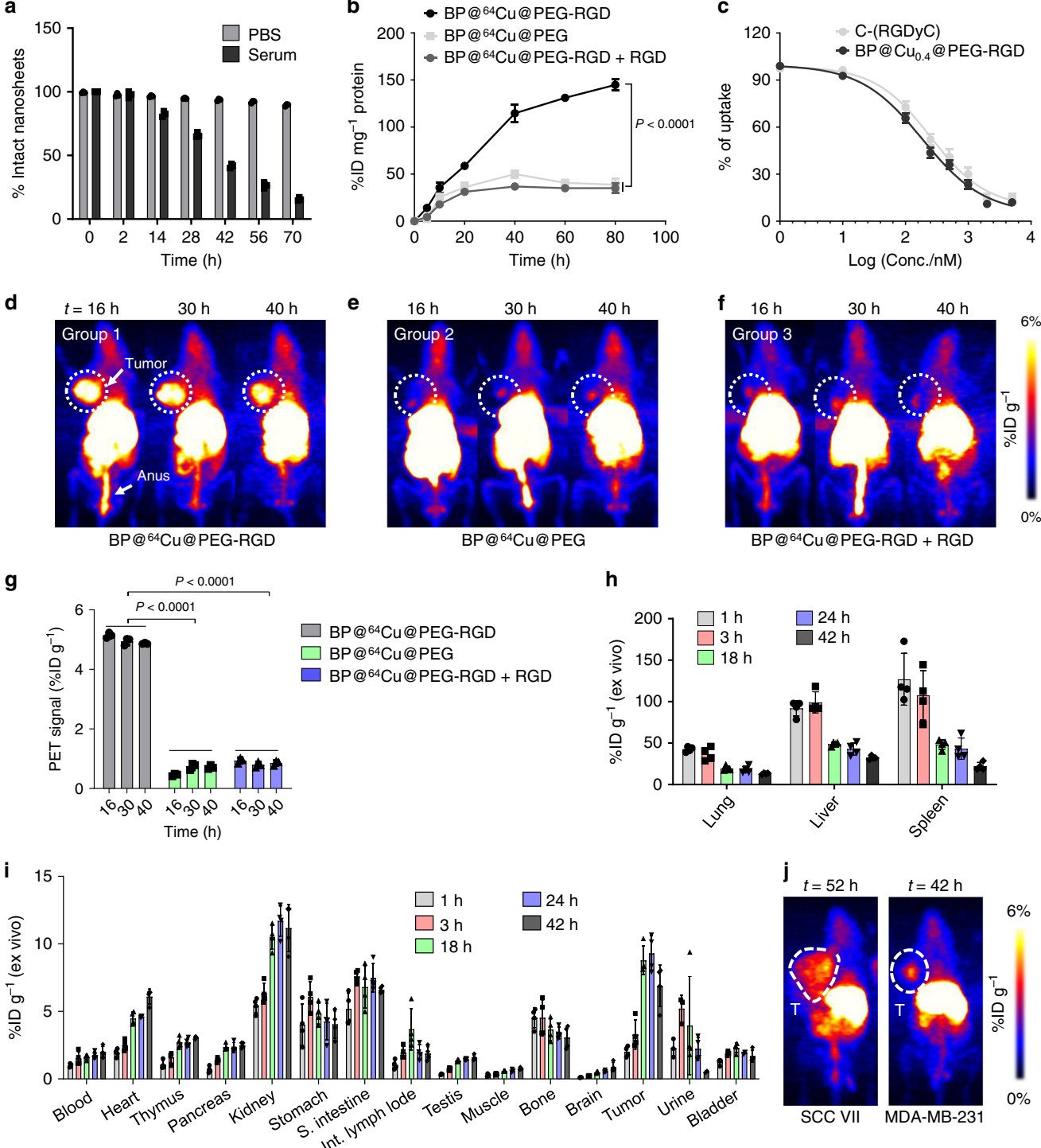

**Fig. 6 Tumor-targeting properties of BP@$^{64}$Cu@PEG-RGD. a** Stability of BP@$^{64}$Cu@PEG-RGD in PBS and mouse serum. Data represents the mean ± s.d., $n = 3$. **b** Time-uptake curves of BP@$^{64}$Cu@PEG-RGD, BP@$^{64}$Cu@PEG, and BP@$^{64}$Cu@PEG-RGD + RGD in B16F10 cells, mean ± s.d., $n = 3$. Analyzed by two-way ANOVA, followed by Tukey's multiple comparisons post-test. **c** Competition binding of BP@$^{64}$Cu@PEG-RGD by various concentrations of c (RGDyC) or BP@Cu@PEG-RGD. The data represents the mean ± s.d., $n = 3$ biologically independent cells. **d–f** MIP PET images of B16F10 tumor-bearing mice at 16, 30, and 40 h after intravenous injection of BP@$^{64}$Cu@PEG-RGD, BP@$^{64}$Cu@PEG, or BP@$^{64}$Cu@PEG-RGD + RGD, respectively. For the BP@$^{64}$Cu@PEG-RGD + RGD group, the c(RGDyC) peptide (5 mg/kg) was co-injected with BP@$^{64}$Cu@PEG-RGD. The white circles denote the tumor sites. **g** Quantification of radioactivity in the tumor from the PET images. The data show mean ± s.d., $n = 3$. Analyzed by two-way ANOVA, followed by Tukey's multiple comparisons post-test. **h, i** Ex vivo biodistribution of BP@$^{64}$Cu@PEG-RGD in tumor and major organs of mice bearing B16F10 tumor at 1, 3, 18, 24, and 42 h p.i. Each point corresponds to mean ± s.d., $n = 4$. **j** MIP PET images of BP@$^{64}$Cu@PEG-RGD in SCC VII and MDA-MB-231 tumor-bearing mice 52 or 42 h p.i., respectively. The white circles denote the tumor sites. All error bars in this figure indicate standard deviation.

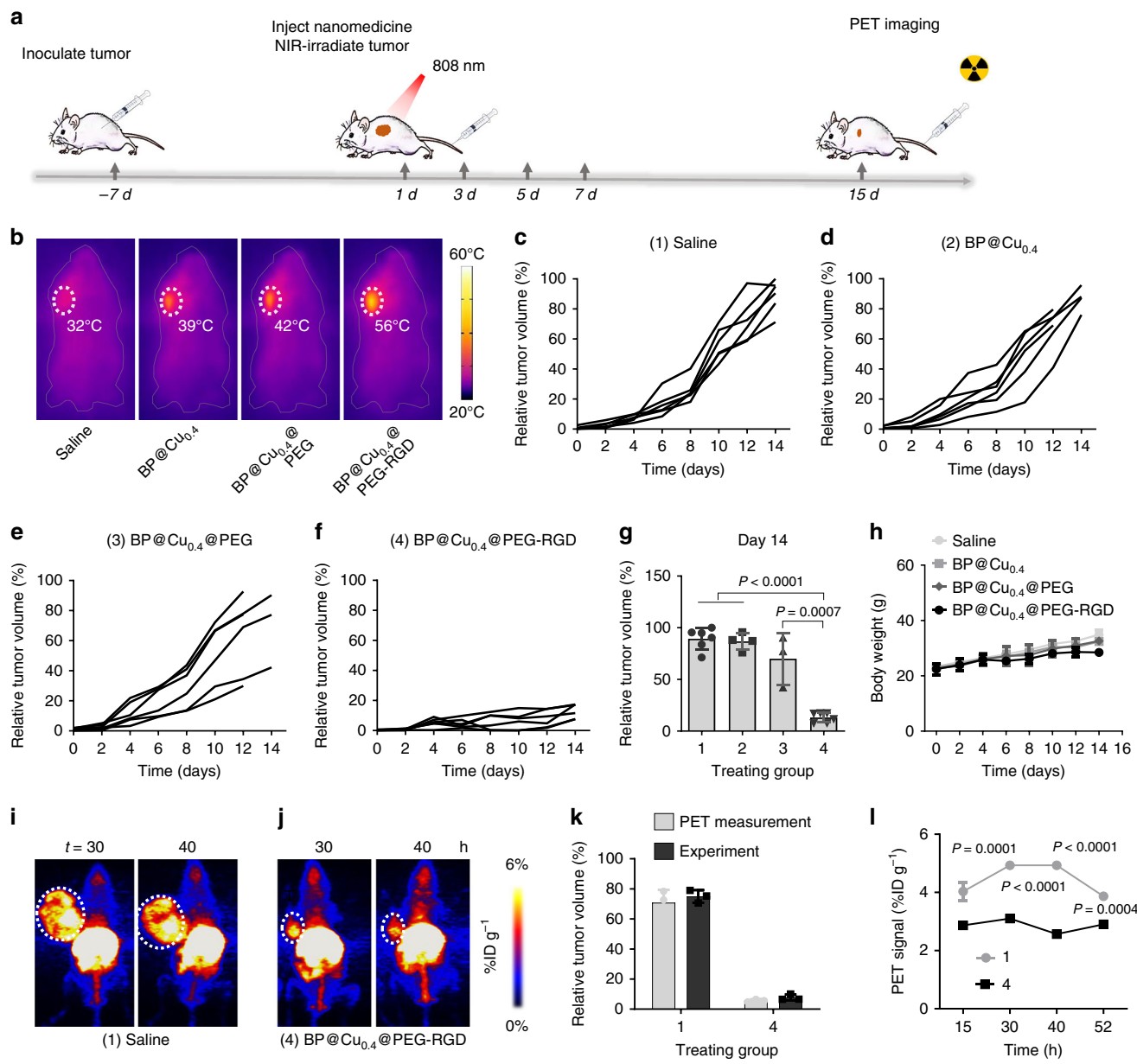

**Fig. 7 In vivo tumor therapy. a** Therapy approach for tumor-bearing mice. **b** Infrared thermal images of tumor-bearing mice injected with (1) saline, (2) BP@Cu$_{0.4}$, (3) BP@Cu$_{0.4}$@PEG, or (4) BP@Cu$_{0.4}$@PEG-RGD under 808 nm laser irradiation (1.0 W cm$^{-2}$) for 2 min. The white circles denote the tumor sites. **c–f** B16F10 tumor growth curves for each mouse after different treatments: **c** saline, **d** BP@Cu$_{0.4}$, **e** BP@Cu$_{0.4}$@PEG, or **f** BP@Cu$_{0.4}$@PEG-RGD. **g** Relative tumor volume after different treatments on day 14. mean ± s.d., $n = 4$–6 biologically independent mice. The number 1–4 was assigned different treatments indicated in (**c**-**f**). The same below. The statistical significance was assessed using unpaired two-tailed student's $t$-test. **h** Body weight curves of B16F10 tumor-bearing mice after different treatments, mean ± s.d., $n = 6$. **i, j** MIP PET images of B16F10 tumor-bearing mice after 2 weeks of treatment with saline or BP@Cu$_{0.4}$@PEG-RGD, respectively. PET images were taken at 30 and 40 h after intravenous injection of saline or BP@$^{64}$Cu@PEG-RGD. The white circles denote the tumor sites. **k** Comparison of tumor volumes from PET images and experimental measurements, mean ± s.d., $n = 3$ biologically independent mice. **l** Quantification of BP@$^{64}$Cu@PEG-RGD uptake in tumors at different time points. The PET signal intensities were analyzed based on PET images in (**i**) and (**j**). The data represents the mean ± s.d., $n = 3$ biologically independent mice. The error bars represent s.d. values. Analyzed by two-way ANOVA, followed by Bonferroni's multiple comparisons test.

significantly larger than those in mice treated with BP@Cu$_{0.4}$@-PEG-RGD (Fig. 7j). From the 3D PET images, we measured the tumor size and compared them with the tumor volumes from experimental measurements. The data from different measurement methods matched each other perfectly, suggesting the feasibility of using PET imaging to plot tumor size and monitor tumor progression (Fig. 7k). Moreover, we used PET to measure the relative uptake of the nanodrugs in the tumor. The mice treated with BP@Cu$_{0.4}$@PEG-RGD demonstrate smaller %ID g$^{-1}$

than that of saline-treated mice, indicating less-aggressive tumors (Fig. 7l). These results suggest that PET imaging is a reliable tool to monitor disease progression, assess therapeutic effects, and even predict remote metastasis, all of which can fundamentally improve treatment planning in clinical practice[70].

To study the toxicity of BP@Cu nanomaterials, we performed hematology analysis, blood biochemical analysis, and histological analysis of major organs of healthy C57/BL6J mice treated by i.v. injection with different agents. No apparent histological

abnormalities and lesions were observed in all the tested organs, suggesting negligible organ toxicity of the BP@Cu nanomaterials (Supplementary Fig. 29). The body weight of mice was monitored for one month following the injection, and no noticeable body weight loss of the mice was observed (Supplementary Fig. 30). Furthermore, blood biochemical analysis showed no great difference between the BP@Cu$_{0.4}$@PEG-RGD and the saline-treated group (Supplementary Fig. 31), further demonstrating the good biocompatibility of our nanomaterial. Finally, the standard hematology markers were measured. Compared with the saline-treated group, all the BPNS nanomaterials treated groups displayed similar results and no statistical significance among them (Supplementary Fig. 32). These results indicate that no obvious infection and inflammation were elicited by the nanomaterials. Taken together, the BP@Cu$_{0.4}$@PEG-RGD display satisfiable in vivo biocompatibility in the mouse model. This lays a good foundation for further clinical translation study of these materials.

## Discussion

High photothermal stability and rapid degradation are two highly desirable properties for a PTA in the clinical application of robust PTT. However, photothermal stability and degradation are usually mutually exclusive in conventional PTAs. That is, PTAs with high photothermal stability usually suffer from slow or difficult degradation, causing potential safety concerns in clinical translation[16]. On the other hand, though PTAs with rapid degradability have shown attractive clinical translation, they usually lack high photothermal stability for effective therapeutic outcomes. Effective strategies to develop an effective PTA with rapid degradability and high photothermal stability have remained elusive.

In this study, we utilized a strategy to make the PTA "coin" show both sides at the same time. This strategy is based on BP PTAs, which have an inherent ability to capture Cu$^{2+}$. Combining BP and Cu$^{2+}$, the developed BP@Cu nanostructures showed enhanced photothermal stability and performance, while their degradation kinetics were robustly accelerated. This strategy addresses the above-referenced PTA conundrum: (1) Cu-based materials are good PTA candidates that can enhance photothermal performance when combined with BP; and (2) Cu$^{2+}$ can accelerate the degradation of BP via redox reaction to achieve rapid degradation. Compared with previous approaches that sacrificed the rapid degradability of BP to enhance photothermal stability[28–31], our intelligent strategy reported here maintains excellent photothermal stability while enabling accelerated degradation. Interestingly, the incorporation of Cu$^{2+}$ further enables BP@Cu nanostructures to produce more cytotoxic hydroxyl radicals for CDT. Moreover, simply by using $^{64}$Cu$^{2+}$, the developed PTA can be also applied in PET imaging for in vivo real-time monitoring of biodistribution and therapeutic outcomes. Therefore, we also achieve the application of BP-based materials in PET-guided, CDT-enhanced combination cancer therapy. Considering that both P and Cu are necessary elements in the human body that play vital roles in human health, this unique biosafety might highlight the clinical potential of our BP@Cu-based PTAs. We expect that this BP@Cu nanoplatform could also be modified with different tumor-targeted ligands and rationally combined with various therapeutic modalities for effective combinatorial cancer treatment.

In summary, the marriage of BP and Cu$^{2+}$ represents a different PTA concept with potential as a unique resolution of the conflict between high photothermal stability and rapid degradability of PTAs. Moreover, considering the outstanding potential of PTT in the clinic and the significant impact of BP in biomedical applications, this study paves the way for the potential application of BP in PET-imaging-guided combination cancer therapy.

## Methods

**Materials and instruments**. BP crystals were commercially obtained from XFNANO Materials Tech Co., Ltd. (Nanjing, China) and stored in an Ar glove box in the dark. Other chemicals were purchased from FUJIFILM Co. Ltd (Tokyo, Japan) or Sigma-Aldrich (St. Louis, MI, USA). All chemical reagents were analytical reagent-grade without further purification. The exfoliated BPNS were stored in N-methyl-2-pyrrolidone (NMP) (99.5%, anhydrous). The $^{64}$Cu was produced in-house at the National Institute of Radiological Sciences (Chiba, Japan) with 98% radionuclidic purity. An 1480 Wizard automatic gamma counter (PerkinElmer, Waltham, MA, USA) was used to measure radioactivity, as expressed in counts of radioactivity per minute (CPM), accumulating in cells and animal tissues. A dose calibrator (IGC-7 Curiemeter; Aloka, Tokyo, Japan) was used for the other radioactivity measurements. The Mal-PEG-NH$_2$ (M.W. = 5000, Catalog No. HEP0203) and FITC-PEG-NH$_2$ (M.W. = 5000, Catalog No. FL044005) were purchased from Advanced BioChemicals. The c(RGDyC) (Catalog No. PCI-3912-PI) was bought from Peptide International, Inc (USA). The PBS, Fetal bovine serum (FBS) and Dulbecco's modified Eagle medium (DMEM) were purchased from Gibco Life Technologies. Cell proliferation kit I (MTT reagent) was purchased from Roche (Basel, Swiss, Catalog No. 11465007001). FITC Annexin V Apoptosis Detection Kit with PI was purchased from Biolegend (San Diego, USA, Catalog No. 640914). Phosphate Assay Kit (Colorimetric, Catalog No. ab65622), Cellular ROS/Superoxide Detection Assay Kit (Catalog No. ab139476), TMRE Mitochondria Membrane Potential Assay Kit (Catalog No. ab113852), Cytochrome C Oxidase Assay Kit (Catalog No. ab239711) and Propidium iodide flow cytometry kit (Catalog No. ab139418) for cell cycle analysis were purchased from Abcam (Cambridge, UK). SYTOX Green Nucleic Acid Stain (Catalog No. S7020), APO-BrdU TUNEL Assay Kit (Catalog No. A23210), Vybrant DyeCycle Orange stain (Catalog No. V35004), CellEvent Caspase-3/7 Green Detection Reagent (Catalog No. C10423) and Hoechst 33258 for nucleus staining (Catalog No. H3596) was purchased from Invitrogen (CA, USA).

**Preparation of BPNS**. Bulky BP crystal (10 mg) was added to a 100 mL sealed flask with 40 mL of NMP. Then the solution was sonicated by a pin sonicator for 24 h at a power of 200 W. The resulting brown dispersion was centrifuged (4 °C) at 4000 rpm for 30 min to remove multilayer BPNS and the supernatant containing BPNS was carefully collected. The collected BPNS were centrifuged at 14,000 rpm for 40 min to remove NMP and washed with ultrapure water twice. To obtain BPNS of consistent size, the resuspended BPNS solution was filtered through a Polyester Membrane Filter (0.2 μm). Finally, the BPNS were resuspended in ultrapure water as a brown solution for further use and stored at 4 °C.

**Characterization of BPNS**. The morphology and size of BPNS were characterized by TEM (HITACHI, HT7700, 100 kV, Japan) and AFM (Bruker, USA). The hydrodynamic size distribution of BPNS was acquired using a Malvern Zetasizer Nano-ZS instrument (ZEN3600, Malvern Instruments, UK) equipped with a temperature controller. A Horiba Jobin – Yvon Lab Ram HR VIS high-resolution confocal Raman microscope was used to obtain the confocal Raman scattering spectra (excitation wavelength: 633 nm). The XPS spectrum was recorded on an X-ray photoelectron spectroscopy (ESCALAB 250, Thermo Fisher, USA). A UV-Vis spectrophotometer (The Agilent Cary 8454 UV-Visible spectrophotometer, CA, USA) was used to measure the optical absorbance of BPNS species. The infrared thermographic and changes in temperatures of the tumor were recorded by a FLIR E6 infrared camera (Arlington, VA). The EPR data was acquired in a Bruker EMXPlus-10/12 system (MA, USA).

**Preparation of BP@Cu@PEG-RGD**. The Cu$^{2+}$ ion stock solution was prepared from the metal salts CuSO$_4$·5H$_2$O. The BPNS aqueous solution was mixed with the metal ion stock solution in the designated ratio. After vigorous stirring for 10 min, the BP@Cu hybrid nanostructure was gathered by centrifugation at 10,000 rpm and the precipitates were washed with water twice. Afterward, the BP@Cu was resuspended in degassed water for storage and further use. For surface modification, Mal-PEG-NH$_2$ (1 equivalent, 10 mg mL$^{-1}$) dissolved in DMSO/H$_2$O solution (1:1 by volume) was reacted with c(RGDyC) (1.2 equiv.) peptide dissolved in DMSO. The mixture was stirred vigorously for 6 h and subjected to further use without purification. The NH$_2$-PEG-RGD (~1 mg) was mixed with the as-prepared BP@Cu nanosheets in aqueous solution under continuous gentle agitation. The pH value was adjusted to ~8.0 with a 0.1 M ammonia aqueous solution. The mixture was gently shaken for 6 h at room temperature to guarantee robust binding, and then the PEGylated BPNS were enriched by centrifugation and refined in PBS.

**NIR laser-induced photothermal conversion**. The prepared nanomaterials were dispersed in water at proper concentrations and then exposed to 808 nm laser irradiation with a fiber-coupled continuous semiconductor diode laser (Beijing Viasho Technology Co., Ltd., China). The laser's power density is 1 W cm$^{-2}$. The

entire surface of the sample was covered by the laser to ensure homogeneous heating. During the irradiation course, an IR thermal camera was employed to monitor the temperature variations of the samples. Based on the previous report, the PTCE ($\eta$) was calculated based on Eq. (1):

$$\eta = \frac{hS\Delta T_{max} - Q_s}{I(1 - 10^{-A_{808}})} \quad (1)$$

where $h$ means the heat transfer coefficient of the solvent, S represents the surface area of the container, $\Delta T_{max}$ is the maximum temperature variation during the heating course. $Q_S$ is the dissipating heat by the centrifuge tube containing solvent after absorbing light, determined independently using a centrifuge tube filled with the solvent. $I$ means the laser power density, and $A_{808}$ refers to the absorbance of the sample at 808 nm. The value of $hS$ is obtained from Eq. (2):

$$hS = mc/\tau \quad (2)$$

where m refers to the solution mass, c means the specific heat capacity of the solvent, and $\tau$ is obtained from the corresponding time of the cooling process.

**Phosphate release assay.** The degradation of BPNS in water was detected by the concentrations of phosphate anions in the supernatant with the Phosphate Assay Kit (Abcam, ab65622) according to the manufacturer's protocol. Briefly, the standard curve for the concentration of phosphate anion was prepared, the phosphate reagent was added to BPNS or BP@Cu samples, and then the optical density was measured by a microplate reader (Bio-Rad Mode 680, USA) at 650 nm (OD650 nm). The data were processed by GraphPad Prism 8.0 software (GraphPad Software, La Jolla, CA, USA).

**Cell Culture.** All cell lines used were purchased from American Type Culture Collection (ATCC). Human breast adenocarcinoma (MDA-MB-231), murine squamous carcinoma (SCC VII), Uppsala 87 Malignant Glioma U87MG, and mouse melanoma (B16F10) cells were cultured in DMEM medium supplemented with 10% FBS, 1% streptomycin/penicillin. All cells were cultured and maintained in a standard cell incubator (37 °C, 5% CO$_2$).

**Cytotoxicity assay.** The cytotoxicity of BP-based nanomaterials was determined by MTT assay. Cells were seeded on 96-well plates at a density of $5 \times 10^3$ cells per 100 µL and cultured for 12 h. Then the medium was replaced with 100 µL of fresh medium containing different concentrations of BPNS, BP@Cu$_{0.2}$, BP@Cu$_{0.2}$@PEG, BP@Cu$_{0.2}$@PEG-RGD, or BP@Cu$_{0.4}$@PEG-RGD. After incubation for 48 h, MTT labeling reagent (10 µL per well) was added and incubated in a cell incubator for 4 h. Then the solubilization solution (100 µL per well) was added and incubated in a cell incubator overnight. Three parallel wells were set for every sample. The absorbance was measured by a microplate reader (Bio-Rad Mode 680, USA) at 590 nm (OD590 nm) with a reference wavelength of 690 nm (OD690nm). The following Eq. (3) was used to calculated the cell viability:

$$
\begin{aligned}
\text{Cell viability (\%)} = &(OD_{590nm}/\text{sample} - OD_{690nm}/\text{sample})/ \\
&(OD_{590nm}/\text{control} - OD_{690nm}/\text{control}) \times 100\%
\end{aligned} \quad (3)
$$

**Apoptosis detection assay.** For quantitative analysis of apoptosis-mediated cell death, B16F10 melanoma cells were seeded in 12-well plates ($10^5$ cells per well) and cultured overnight. The cells were co-incubated with different BPNS species for 24 h and a NIR laser was applied at 4-h post-incubation for 2 min. Then the cells were trypsinized, washed, and resuspended in ice-cold PBS. Apoptotic cells were identified with a FITC Annexin V/PI Apoptosis Detection Kit (Biolegend, 640914) according to the manufacturer's protocol. Briefly, the cells were washed and resuspended in 100 µL of binding buffer (1×) containing 5 µL of PI and 5 µL of Annexin V-FITC. Subsequently, the solution was incubated in the dark for 15 min at room temperature. Afterward, the cells were analyzed by flow cytometer (FACSCalibur, BD). The data were further processing using the Flowjo software (BD Biosciences, New Jersey, USA). The experiments were done triplicates to obtain reliable data.

**Cell cycle detection assay.** B16F10 cells were seeded at a density of $10^5$ cells per well on 12-well plates and cultured overnight. The cells were co-incubated with different BPNS species for 24 h and a NIR laser was applied at 4-h post-incubation for 2 min. After incubation, the B16F10 cells were harvested, washed, and resuspended with ice-cold PBS. Then cell cycles were determined with the Propidium Iodide Flow Cytometry Kit (Abcam, ab139418) based on the manufacturer's protocol. Briefly, the harvested cells were fixed in 66% ethanol at 4 °C overnight and then rehydrated in PBS. The cells were stained with PI and RNase for 30 min at room temperature in the dark. Afterward, the DNA contents were analyzed by the flow cytometer (FACSCalibur, BD). The data were analyzed by Flowjo software (BD Biosciences, New Jersey, USA). The experiments were done triplicates to obtain reliable data.

**ROS/superoxide detection assay.** B16F10 cells were seeded in 12-well plates at a density of $10^5$ cells per well to ensure ~50–70% confluency on the day of the experiment. This assay was performed according to the manufacturer's protocol for the cellular ROS/superoxide detection assay kit (Abcam, ab139476). Briefly, before treatment, the medium was replaced with fresh medium. Then, either vehicle or BPNS species was added to a desirable working concentration and incubated for 120 min in a cell incubator, at the end of incubation, NIR laser was applied for 2 min. Then the cells were harvested and washed with ice-cold PBS. The cells were further treated with 500 µl of ROS/Superoxide Detection Solution and incubated for 30 min at 37 °C in the dark. The cells were then analyzed by flow cytometer (FACSCalibur, BD) and the data analyzed using Flowjo software (BD Biosciences, New Jersey, USA). Three independent experiments were performed to obtain reliable data.

**Animal models and tumor inoculation.** Male BALB/c and BALB/c slc nu/nu mice, C3H/HeJ mice, and C57BL/6J Jms mice (all 6-weeks old) were purchased from Japan SLC (Shizuoka, Japan). All animals received humane care, and the Animal Ethics Committee of the National Institute of Radiological Sciences approved all the animal experiments. All experiments were carried out according to the recommendations of the Committee for the Care and Use of Laboratory Animals, National Institute of Radiological Sciences. The laboratory mice were housed in the ABSL-3 lab where were maintained under constant conditions (21 ± 1 °C temperature; 50 ± 5% humidity; −140 ± 10 pa negative air pressure; <55 dB background noise) on a 12 h:12 h light:dark cycle. For the B16F10 tumor model, $5 \times 10^5$ cancer cells suspended in 100 µl DMEM medium were injected subcutaneously into C57BL/6 J Jms mice or Balb/c mice. For MDA-MB-231 tumor xenograft, $1 \times 10^7$ cells suspended in 100 µl of DMEM medium were subcutaneously implanted into the left foreleg armpit of the BALB/c slc nu/nu mice. For the SCC VI tumor model, $1 \times 10^6$ cells suspended in 100 µl DMEM medium were subcutaneously implanted into the left foreleg armpit of C3H/HeJ mice. When the tumor diameters reached 3–5 mm, the tumor-bearing mice were used for further studies.

**$^{64}$Cu labeling method.** The BPNS nanomaterials were suspended in water (0.1 mg per 100 µL). The refined $^{64}$CuCl$_2$ solid was dissolved in sodium acetate buffer (pH = 4.1) to a final concentration of 10–20 mCi per100 µL. Then 100 µL of BPNS solution was mixed with 100 µL $^{64}$CuCl$_2$ solution at room temperature. The mixture was gently sonicated for 10 min at a power density of 50 W. Thin-layer chromatography (TLC) using 0.05 M ethylenediaminetetraacetic acid (EDTA) as the mobile phase was used to check the labeling yields. Afterward, the BP@Cu hybrid nanomaterials were collected by centrifugation for 20 min at 12,000 rpm to remove the free $^{64}$Cu. The precipitate was washed in water and then resuspended in water for further use.

**Labeling efficiency.** Assessment of the stability of BP@$^{64}$Cu@PEG-RGD in saline and in mouse serum was carried out based on the following protocol or using radio-TLC (see in supplementary materials). The BP@$^{64}$Cu@PEG-RGD solution (3.7 MBq) was added to 90 µl of mouse serum (freshly prepared) or 90 µL of PBS and then incubated at 37 °C for a designated time. Aliquots of the mixture were removed at designated time points and 100 µL of acetonitrile and water (1:1, v/v) was then added. The suspension was subjected to centrifugation for 10 min at 10,000 rpm. The supernatant was decanted and the radioactivity in supernatant and precipitant were measured using a gamma counter. The percentage of retained $^{64}$Cu in the BP@$^{64}$Cu@PEG-RGD nanomaterials were calculated using the following formula: (total radioactivity − radioactivity in filtrate)/total radioactivity.

**Cellular uptake assay.** B16F10 cells seeded in 24-well plates ($5 \times 10^4$ cells per well) were maintained for 24 h in a humidified atmosphere. Then the medium was removed, and the cells were washed three times with PBS. The medium containing radionanomaterials (740 KBq mL$^{-1}$) was added to each well. The plates were incubated at 37 °C for designated periods (5, 10, 20, 40, 60, and 80 min); the medium was removed from three wells at each time point, and the cells were washed three times with PBS and lysed with 0.5 mL NaOH (0.1 mol L$^{-1}$). The lysate was harvested and its radioactivity measured with the automatic gamma counter. Decay corrections were performed for all radioactivity measurements.

**PET imaging assay.** PET scans were conducted using an Inveon PET scanner (Siemens Medical Solutions, Knoxville, TN, USA), which provides 159 transaxial slices (Center-to-center spacing: 0.796-mm; transaxial field of view: 10 cm; and axial field of view: 12.7 cm). During the scan, all mice were anesthetized with 1–2% (v/v) isoflurane and kept in a prone position. The BP@$^{64}$Cu@PEG-RGD or BP@$^{64}$Cu@PEG (10–17 MBq per 100 µL) was injected via a preinstalled tail vein catheter. Immediately after injection, a dynamic scan in 3D list mode was acquired at different time points. For each type of tumor, at least three mice were imaged, and MIP images were obtained for all mice. PET static images were reconstructed using an analysis software (ASIPro VM, Siemens Medical Solutions). A filtered back-projection using Hanning's filter with a Nyquist Cutoff of 0.5 cycle/pixel was used to reconstruct the images. Volumes of interest areas were analyzed using ASIPro software. All radioactivity was decay-corrected based on the injection time and expressed as the percent of the total injection dose/per gram tissue (%ID g$^{-1}$).

For the blocking PET study, excess free c(RGDyC) peptides (5 mg kg$^{-1}$) were coinjected with BP@$^{64}$Cu@PEG-RGD.

**Ex vivo biodistribution assay**. The BP@$^{64}$Cu@PEG-RGD dissolved in saline (3.7 MBq per 100 μL) was injected into tumor-bearing C57BL/6J Jms mice via the tail vein. The mice ($n = 4$ for each time point) were sacrificed by cervical dislocation at designated time points postinjection. The tumors and major organs (including heart, liver, lung, spleen, pancreas, kidneys, stomach, muscle, small intestine, intestinal lymph node, testis, muscle, bone, brain, and blood) were quickly harvested and weighed. The radioactivity in these organs was measured using the automatic gamma counter. The results are expressed as the percentage of injected dose per gram of wet tissue (% ID/g). Decay corrections were performed for all radioactivity measurements.

**In vivo tumor inhibition assay**. B16F10 cells ($5 \times 10^5$) in 100 μL of DMEM medium were subcutaneously inoculated into the flank of 6-week-old C57BL/6J Jms mice. About one week later, when the tumor volume grew to about 50 mm³, the mice were used for the therapy study. For tumor therapy by intratumoral injection, the mice were randomly divided into four treatment groups (five mice per group) and treated as follows: (I) saline, (II) BPNS, (III) BP@Cu + NIR. Saline (50 μL) or BPNS species (2 mg BPNS mL$^{-1}$ in saline, 50 μL per mouse) was intratumorally injected into mice. For group 4, an NIR laser (1 W cm$^{-2}$) was applied to irradiate the entire tumor for 2 min after injection for 4 h. Four injections were applied in total. For tumor therapy by i.v., the mice were randomly divided into four treatment groups (six mice per group) and the treatment settings were as follows: (I) saline, (II) BP@Cu, (III) BP@Cu@PEG, (IV) BP@Cu@PEG-RGD. Saline (100 μL) or BPNS species (1 mg BPNS mL$^{-1}$ in saline, 100 μL per mouse) was intravenously injected t. Twenty-four hours after injection, the tumor of mice in all groups were exposed to 808 nm light (1 W cm$^{-2}$, 5 min). The surface temperature variations of the tumor were recorded by an infrared thermal camera. The tumor sizes were measured every 2 days using a digital Vernier caliper. The body weights of mice were weighed using an electronic scale, respectively. The following formula was used to estimate the tumor volume: (tumor length) × (tumor width)$^2$/2. The final data are presented as relative tumor volume, which is defined as the ratio of the tumor volume at the measurement day against the maximum tumor volume of all the mice in the group on day 14.

**Statistical analysis**. Graph Pad Prism 8.0 and Origin 9.0 were used for data statistics and statistical significance calculation. Microsoft Excel 2016 was used for biodistribution and tumor size analysis. PET images were analyzed using ASIPro VM version 6.3.3.0 software (SiemensMedical Solutions). Statistical analysis was performed using the one way/two-way ANOVA or the Student's t-test with statistical significance assigned at ***$P < 0.05$ (significant), **$P < 0.01$ (moderately significant), ***$P < 0.001$ (highly significant) and ****$P < 0.0001$ (extremely significant).

**Reporting summary**. Further information on research design is available in the Nature Research Reporting Summary linked to this article.

## Data availability

The authors declare that all data supporting the findings of this study are available within the article and the Supplementary Information. Other data are available from the corresponding authors upon reasonable request.

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

## Acknowledgements

We thank the staff of the National Institutes for Quantum and Radiological Sciences and Technology (QST) for their help with cyclotron operation, radioisotope production, radiosynthesis, and animal experiments. We are sincerely grateful for the financial support of the Ministry of Education, Culture, Sports, Science and Technology of the Japanese Government (Basic Research B: No. 17H04267, M.-R.Z.; Grant-in-Aid for Young Scientists: No. 19K17156, K.H.; Grant-in-Aid for Research Activity Start-up: No. 18H06217, K.H.), National Institutes for Quantum and Radiological Science and Technology of Japan (Granted FY2019 Diversity Promotion Collaborative Research Subsidy, Presidential strategic fund for budding research, K.H.), U.S. METAvivor Early Career Investigator Award (No. 2018A020560, W.T.), Harvard Medical School (HMS)/ Brigham and Women's Hospital (BWH) Department of Anesthesiology-Basic Scientist Grant (No. 2420 BPA075, W.T.), Stepping Strong Breakthrough Innovator Award, the National Natural Science Foundation of China (81701751, L.W.; 81871383, H.X.) and the Science and Technology Program of Guangzhou (201804010440, L.W.).

## Author contributions

K.H., W.T., and M.R.Z. conceived and designed the project. K.H., L.W., J.W., M.H., and J.O. synthesized and characterized the materials. K.H., L.X., Y.D.Z., X.Y.J., and Z.M.Y. performed all the in vitro and in vivo experiments. M.H., K.N., and H.S. produced the ⁶⁴Cu. K.H., O.C.F., H.X., S.H.L., L.W., W.T., and M.R.Z. discussed the results and interpreted the data. K.H., W.T. and M.R.Z. co-wrote the paper, and all authors commented on and approved it.

## Competing interests

The authors declare the following competing financial interests: O.C.F. has financial interests in Selecta Biosciences, Tarveda Therapeutics, and Seer. The remaining authors declare no competing interests.
