## [Peer Review File · Nature Communications]

Reviewers' Comments:

Reviewer #1:

Remarks to the Author:

Hu et al. reported the coordination between black phosphorus and copper ions to achieve combination cancer theranostics. Although interesting and solid results were shown in the manuscript, there are still some flaws and deficiencies that should be improved in the manuscript.

1. Introduction part, there are a great variety of nanoplatforms that reported the coordination between nanoparticles and copper ions to achieve combination cancer theranostics (ACS Nano. 2019,13:4267, 2017, 11:9103; Nano Lett. 2016, 16: 5601; ACS Nano.; Biomaterials. 2015, 57:41, et al.), please refer these recent articles and reviews.
2. BP@Cu nanotheranostics could be used as photoacoustic imaging agents, no need to label the $^{64}\text{Cu}^{2+}$. please discuss and explain why not choose the simple imaging method.
3. The inherent Cu^{2+} -capture ability of black phosphorus can accelerate the degradation of black phosphorus via a redox reaction, please discuss and explain how to determine the free $^{64}\text{Cu}^{2+}$ and the labeled black phosphorus in PET images.
4. For abbreviation, the authors should use the same name throughout the manuscript and in Fig 7, it is not clear why H is described twice.

Reviewer #2:

Remarks to the Author:

Kuan and co-workers has presented a nice piece of work about BP-based materials in PET-guided, CDT-enhanced combination cancer therapy. The manuscript show detailed material characterization and in vitro and in vivo studies. However a few experiments and explanation are necessary before it meet the publication standard. In general author should give more detail explanation of experiments and data processing.

- Author claims that BPNS undergo redox reaction in tumor microenvironment. However there is no direct proof of the Redox reaction happening in the context. The literature report indicating the Fenton like agent is not exactly identical of the formulation of current manuscript.
- TEM and HRTEM images of BP@Cu and BP@Cu@PEG-RGD will give direct and accurate evidence of the structural integrity of the nanosheet structure.
- According to the design and Fig-1 the material contains both Cu^{+2} and Cu^{+1} . However Cu^{+1} is not a stable agent in the biological environment. Specific coordination complexes, such as TBTA, has ben used to stabilize the +1 oxidation state of the Cu in water. Is there any proof of the Cu^{+1} state in the material. EPR spectroscopy could be an experimental tool to check the oxidation state.
- Cyclic peptide c(RGDyC) is conjugated to the PEG molecules by thiol-maleimide chemistry. However in general term the click reaction refers the reaction between alkyne and azide. Here it is better to call as thiol-maleimide conjugation than Click reaction.
- Fig 3a: The absorption spectra of BP@Cux display an interesting outcome. Author should make a clear interpretation of the fig 3a. The BP@Cu0.1 shows a peak at 800nm, which could be the indication of the better PTA upon irradiation of NIR light. Why it has been recorded after 4h of storage in water? Its not clear what is the aim of the experiment, does it correspond to the photochemical property of the material or degradation of that? Absorption spectra at different time points to correlate the coordination of Cu to BP and degradation will ban interesting topic to discuss.
- It is clear that the material degrade with time. What is the rate of degradation in presence of blood or plasma?

- Fig 3b: It is not clear that the difference in amount of phosphate ion released for each type of BP@Cu is significant in all the timepoints?
- Fig 3f-g: A distinguishable difference in the IT maps can be found in case of BP, BP@Cu0.2, and BP@Cu0.2@PEG-RGD. However, in case of Fig 3f both are almost similar. It will be nice to represent the data of Fig 3f with error bar with appropriate statistical test.
- Figure 4a-c: Why the incubation time is different for the 3a and 3b-c? Scale bar is absent in Fig 3b. Appropriate statistical analysis should be performed for Fig 4c.
- Fig 4f has no BP@Cu@PEG-RGD but it is present in fig. 4g. What was the actual experiment and how the fig 4g has been generated?
- No difference between BP@Cu, and BP@Cu@PEG-RGD in case of cell viability (fig 4a,c) but in case of fig 5a the difference is significant. Can author explain this observation?
- Propidium iodide (PI) is the commonly used fluorochrome for the cell cycle analysis because of the optimal linear DNA-binding capacity i.e. they bind in proportion to the amount of DNA present in the cell. The membrane penetration plays a crucial role in PI staining. However, the membrane integrity is significantly different in live/early apoptotic cell with dead/late apoptotic cell. Hence, when cells are undergoing through apoptotic stress, the higher intensity of PI can also signify the late apoptotic cells. So, apoptosis can also play a major role in the increase of the cell population with higher PI stain in the data. Author can use some other cell cycle analysis dyes (such as, DyeCycle dyes) to check the cell cycle distribution in case of BP@Cu. The other way of overcoming this is to perform fixation and permeabilization to have an easy access inside the nucleus.
- Author should give additional evidence for apoptosis, such as, monitoring Caspase or other apoptotic proteins.
- As author claims the apoptosis is initiated by the excess production of the ROS, the involvement of mitochondria can give additional value in the study. The change in mitochondrial membrane potential or Cytochrome-c release can be monitored easily.
- It is not very clear that why the BP@Cu induce apoptosis. To be more specific, why it induce excess production of ROS? Some experimental evidence or explanation should be required in this section.
- Fig 6a: author should provide detailed experimental condition and normalization method for this data.
- Author should provide some direct evidence of the uptake of the BP@Cu-RGD. The uptake data presented in the manuscript can also come from the accumulation of BP@Cu on extracellular membrane. A fluorophore conjugated RGD or PEG on the BP@Cu will give a direct evidence of the cellular uptake.
- In case of biodistribution assay the nanoparticles were injected through tail vein. However the % ID/g in blood remains low at initial time points and then gradually increases with time. Author should explain this observation.
- Author has mentioned in the introduction section that the traditional PTAs are related with the kidney and liver problem and the current material will overcome of that problem. According to the data the BP@Cus has high accumulation in Kidney and liver. Does it create similar toxicity? Is it possible to give some evidence about the hepatic and renal toxicity? Additionally, nanostructures also accumulate in the lungs and spleen (example Graphenes, CNTs). The authors should look at

accumulation and toxicity to these organs.

March 17, 2020

Dear reviewers,

We sincerely appreciate your thoughtful and professional comments on our manuscript. We have adopted virtually all of the suggestions and revised the manuscript accordingly. We hope that the revised manuscript has been improved to the level of the reviewers' satisfaction. We also hope that you will now find it suitable for publication in *Nature Communications*.

Reviewers' comments:

Reviewer #1 (Remarks to the Author):

Hu et al. reported the coordination between black phosphorus and copper ions to achieve combination cancer theranostics. Although interesting and solid results were shown in the manuscript, there are still some flaws and deficiencies that should be improved in the manuscript.

Response: Thanks for the reviewer's positive comments. We have followed the reviewer's comments and performed additional experiments to address the flaws and deficiencies raised by the reviewer. Please see the following responses for more details.

1. Introduction part, there are a great variety of nanoplatforms that reported the coordination between nanoparticles and copper ions to achieve combination cancer theranostics (ACS Nano. 2019,13:4267, 2017, 11:9103; Nano Lett. 2016, 16: 5601; ACS Nano.; Biomaterials. 2015, 57:41, et al.), please refer these recent articles and reviews.

Response: We thank for the comment. In the revised manuscript, we have cited more related references in the "Introduction" section. The newly added references are assigned to ref. no. 34, 35, and 41-45.

34. Liu S, Wang L, Lin M, Wang D, Song Z, Li S, *et al.* Cu(II)-Doped Polydopamine-Coated Gold Nanorods for Tumor Theranostics. *ACS Appl. Mater. Inter.* **9**, 44293-44306 (2017)
35. Liu C, Wang D, Zhang S, Cheng Y, Yang F, Xing Y, *et al.* Biodegradable Biomimic Copper/Manganese Silicate Nanospheres for Chemodynamic/Photodynamic Synergistic Therapy with Simultaneous Glutathione Depletion and Hypoxia Relief. *ACS Nano* **13**, 4267-4277 (2019)
41. Shen S, Jiang D, Cheng L, Chao Y, Nie K, Dong Z, *et al.* Renal-Clearable Ultrasmall Coordination Polymer Nanodots for Chelator-Free ⁶⁴Cu-Labeling and Imaging-Guided Enhanced Radiotherapy of Cancer. *ACS Nano* **11**, 9103-9111 (2017)
42. Liu TW, MacDonald TD, Shi J, Wilson BC, Zheng G. Intrinsically Copper-64-Labeled Organic Nanoparticles as Radiotracers. *Angew. Chem. Int. Ed.* **51**, 13128-13131 (2012)
43. Zhou M, Chen Y, Adachi M, Wen X, Erwin B, Mawlawi O, *et al.* Single agent nanoparticle for radiotherapy and radio-photothermal therapy in anaplastic thyroid cancer. *Biomaterials* **57**, 41-49 (2015)

44. Fan Y, Zhang J, Shi M, Li D, Lu C, Cao X, *et al.* Poly(amidoamine) Dendrimer-Coordinated Copper(II) Complexes as a Theranostic NanoplatforM for the Radiotherapy-Enhanced Magnetic Resonance Imaging and Chemotherapy of Tumors and Tumor Metastasis. *Nano Lett.* **19**, 1216-1226 (2019)
45. Shaffer TM, Harmsen S, Khwaja E, Kircher MF, Drain CM, Grimm J. Stable Radiolabeling of Sulfur-Functionalized Silica Nanoparticles with Copper-64. *Nano Lett.* **16**, 5601-5604 (2016)

2. BP@Cu nanotheranostics could be used as photoacoustic imaging agents, no need to label the $^{64}\text{Cu}^{2+}$. please discuss and explain why not choose the simple imaging method.

Response: Thank you for this important comment. We agree with your comment that the BP@Cu nanotheranostics could be used as photoacoustic imaging (PAI) agents. Several previous studies have successfully applied BP-based nanoagents for PAI. For example, in 2016, Li et al. demonstrated that PEGylated BP nanoparticles are suitable agents for PAI. (*Biomaterials*, **2016**, 91, 81-89) In 2017, Yu et al. reported that TiL4-coordinated BPQDs is an efficient contrast agent for in vivo PAI of cancer. (*Small*, **2017**, 13, 1602896). In 2019, Yang et al. prepared BPNS@TA-Mn nanocomposites and demonstrated them presenting high-contrast imaging performance both in MRI and PAI. (*Chem Commun*, **2019**, 55, 850-853) Based on the above references, it is obvious that the BP-based nanoagents are suitable PAI contrast agents.

PAI has become an important tool to study the in vivo behavior of nanoparticles with PA properties. PAI is a newly emerging bioimaging modality that can overcome the drawbacks of ultrasound imaging (e.g. speckle artifacts) and optical imaging (e.g. limited penetration depth). It can generate high spatial resolution images with optical contrast in a region up to 7 cm deep in biological tissues, and provide inherently background-free detection. Despite PAI is noninvasive and easy-access, it still shows some limitations and challenges. Some of the major limitations of PAI include: (1) Like other kinds of traditional optical imaging techniques, the semiquantitative nature of PAI impedes the accurate and comprehensive evaluation of the drug metabolism and pharmacokinetics (DMPK) of the targeted compounds; (2) although PA imaging is a promising technology, whole-body mapping is still a huge challenge in animals and humans; (3) the intensity of PA signals is linearly-correlated with the amount of the contrast agents in targeted organs, to achieve high-quality images of different parts of the body, the administration dose should be usually optimized. (4) The calculation algorithm for the dose is varied, making the data poor reproducibility among different groups.

Compared to PAI, PET imaging is a more mature technique that has been widely used in the clinic to diagnose diseases and act as a reliable tool to aid the evaluation of DMPK of drug candidates under clinical evaluation. PET can provide a real-time, whole-body, dynamic mapping of the distribution and metabolism of the targeted compounds *in vivo*. Moreover, while the administrated dose in PET imaging is determined by the radioactivity, the physical mass of the tracers can be ignored. Thus, PET imaging usually shows excellent reproducibility across different centers, without affecting by pharmacological profiles of the used tracer.

In this study, our purpose is to perform a comprehensive preclinical evaluation of BPNS-based nanosheets labeled by ^{64}Cu as an “ideal” PPT agent. We aim to not only monitor the tumor engagement of the nanomaterials but also to profile the pharmacokinetics and metabolism routes of the nanomaterials. In this case, PET imaging is the most suitable method to acquire such data. Besides, $^{64}\text{Cu}^{2+}$ coordination makes it possible to perform *ex vivo* biodistribution to quantify the nanomaterials accumulated in various organs. These data are significant to estimate the toxic dose of the nanomaterials. Taken together, we integrated $^{64}\text{Cu}^{2+}$ in the system for quantitative PET imaging and DMPK study of BP@Cu@PEG-RGD in mice other than to select PAI for *in vivo* imaging. To

more clearly elucidate our idea, we have added more discussion and explanation in the revised manuscript. Besides, we cited more references to support our claims.

Revised:

The first paragraph under the section of “Pharmacokinetics, biodistribution, and clearance of BP@Cu nanostructures”

The superior therapeutic effects and biocompatibility of BPNS make it an excellent nanomaterial for cancer therapy. However, the detailed *in vivo* pharmacokinetics and pharmacodynamics of BPNS-based nanodrugs have not been fully clarified. Although several studies have tried to elucidate the biodistribution of BPNS-based nanomaterials, optical-based imaging methods, are limited in terms of the accurate and comprehensive evaluation of pharmacokinetics and distribution because of their semiquantitative nature and limited depth of penetration^{29, 47}. Photoacoustic imaging (PAI) is an attractive bioimaging modality that shows advantages on speckle artifacts and penetration depth than ultrasound imaging and optical imaging, respectively⁷⁶. Several previous studies have demonstrated that BP-based nanomaterials are suitable contrast agents for PAI^{32, 77}. However, PAI still shows some limitations and challenges in real-time, quantitative and full-body mapping for the drug metabolism and pharmacokinetics (DMPK) of the targeted compounds⁷⁸. Thus, it is urgent to quantitatively and accurately evaluate the *in vivo* biodistribution and metabolism profile of BPNS to enable clinical applications. Among existing imaging techniques, PET imaging is an excellent tool to noninvasively assess the pharmacological properties and pharmacokinetics of a variety of drug molecules, given its excellent quantitative capability, unlimited signal penetration, and high detection sensitivity. PET imaging has become one of the most important techniques in both preclinical and clinical scenarios^{79, 80}. In this context, it is conceivable to combine PET imaging functionality with BPNS for accurate quantitative tracking of *in vivo* behavior in real-time.

3. The inherent Cu²⁺-capture ability of black phosphorus can accelerate the degradation of black phosphorus via a redox reaction, please discuss and explain how to determine the free ⁶⁴Cu²⁺ and the labeled black phosphorus in PET images.

Response: Thank you for this valuable comment. To answer this comment and make a clear explanation, we performed PET imaging study and *ex vivo* BioD of free ⁶⁴Cu²⁺ and compared them with that of the nanocomplex. The results are consistent with previous reports (*J Nucl Med*, **2006**, 47,1649–1652; *J Nucl Med*, 2014, 55, 812–817). From these results, we noticed that the free ⁶⁴Cu²⁺ shows very different pharmacokinetics and biodistribution profiles with BP@⁶⁴Cu@PEG-RGD. For ⁶⁴Cu²⁺, the highest uptake in tumor (□ 4.5 %ID/g) appeared at 1 hour following intravenous injection, after that, the ⁶⁴Cu²⁺ was gradually decreased in the tumor sites. For comparison, the peak uptake of the BP@⁶⁴Cu@PEG-RGD (□ 10.0 %ID/g) occurred 24 hours postinjection. Aside from the tumor, the distribution in major organs is significantly different. The top four uptake organs for free ⁶⁴Cu²⁺ are liver, small intestine, lung, and kidney. However, for BP@⁶⁴Cu@PEG-RGD, the organs with the highest accumulation of radioactivity are spleen, then followed by liver, lung, and kidney. These data indicated that the PET images in our study exclusively unveiled the DMPK of BP@⁶⁴Cu@PEG-RGD.

Nevertheless, with the oxidative degradation of BPNS, some free ⁶⁴Cu ions would be released from the BPNSs. However, to the best of our knowledge, there is still a shortage of methods or algorithms to distinguish the ⁶⁴Cu signals from signals generated by ⁶⁴Cu labeled compounds. Furthermore, regarding the degradation of BP@⁶⁴Cu@PEG-RGD is slow based on the *in vitro* stability results (*t*_{1/2} is about 35 hours), the signals from ⁶⁴Cu ions would exert limited perturbation to the whole signals. In summary, to determine the free ⁶⁴Cu²⁺ and the labeled BPNS in PET images should be very interesting and important but unrealizable so far. The selective uptake of ⁶⁴Cu²⁺ in some organs while not for BPNS would be a possible point to unlock this problem and this requires

more anatomic studies in the future. In the revised manuscript, we have included the PET images and *ex vivo* BioD data as Supplementary Fig. 24 and 25. Some descriptions were also added to the manuscript.

The description in the manuscript:

To preclude the radioactivity in the tumor from free ^{64}Cu ions, we performed PET study and *ex vivo* biodistribution of free ^{64}Cu ions in the mouse model (Supplementary Fig. 24 and 25). Both experiments revealed a significant difference between $^{64}\text{Cu}^{2+}$ and BP@ ^{64}Cu @PEG-RGD. The PET images of $^{64}\text{Cu}^{2+}$ showed lower tumor uptake than BP@ ^{64}Cu @PEG-RGD. Then an analysis for the biodistribution data suggested that the highest uptake in tumor ($\square 4.5$ %ID/g) for $^{64}\text{Cu}^{2+}$ appeared at 1 hour following intravenous injection, after that, the $^{64}\text{Cu}^{2+}$ was gradually decreased in the tumor sites. For comparison, the peak uptake of the BP@ ^{64}Cu @PEG-RGD ($\square 10.0$ %ID/g) occurred 24 hours postinjection. Aside from the tumor, the distribution in major organs is significantly different. The top four uptake organs for free $^{64}\text{Cu}^{2+}$ are liver, small intestine, lung, and kidney. These data indicated that the radioactivity in the tumor is exclusively originated from BP@ ^{64}Cu @PEG-RGD rather than free $^{64}\text{Cu}^{2+}$.

Supplementary Fig. 24 MIP PET images of B16F10 tumor-bearing mice at 1, 2, 5, 27, and 42 hours after intravenous injection of free $^{64}\text{Cu}^{2+}$.

Supplementary Fig. 25 *Ex vivo* biodistribution of free $^{64}\text{Cu}^{2+}$ ions in tumor and major organs of mice bearing B16F10 tumor at 1, 3, and 18 hours postinjection. Each point corresponds to mean \pm s.d., $n = 3$.

4. For abbreviation, the authors should use the same name throughout the manuscript and in Fig 7, it is not clear why H is described twice.

Response: Thanks for careful reading. We unified the name of the compounds throughout the manuscript. For Fig. 7H, we apologize to make this mistake. We have modified the figure 7 and the corresponding description in the manuscript text.

Reviewer #2 (Remarks to the Author):

Kuan and co-workers has presented a nice piece of work about BP-based materials in PET-guided, CDT-enhanced combination cancer therapy. The manuscript show detailed material characterization and in vitro and in vivo studies. However a few experiments and explanation are necessary before it meet the publication standard. In general author should give more detail explanation of experiments and data processing.

Response: Thank you for the nice comments. During the past several months, we have performed more experiments to acquire more data. All these data are either added to the manuscript or the supplementary material. Moreover, based on your comments, we have done a series of modifications/corrections/additions to the manuscript. We wish that the revised version will satisfy the high publication standard of the journal and the reviewer.

• *Author claims that BPNS undergo redox reaction in tumor microenvironment. However there is no direct proof of the Redox reaction happening in the context. The literature report indicating the Fenton like agent is not exactly identical of the formulation of current manuscript.*

Response: Thank you for this insightful comment. As you said, the exact form of copper ion for Fenton like reaction is cuprous (Cu^+). However, because of the cuprous is very unstable, people usually use divalent copper ions (Cu^{2+}) as a precursor form for Fenton like reaction. The logic is that Cu^{2+} can react with GSH in TME and itself is reduced to Cu^+ . This kind of design has several advantages. First, it overcomes the instability issue of Cu^+ . Second, Cu^{2+} can deplete the GSH in TME, thus relieve the tumor antioxidant ability and weaken the capacity of tumor cells to scavenge ROS. Third, the Cu^{2+} shows little toxicity to normal cells and only elicits Fenton like reaction in malignant cells because of the high level of GSH in tumor cells. In our system, the core idea is depicted as follows. The Cu^{2+} captured by BPNS can react with BPNS via a redox reaction to generate Cu^+ . After that, the generated Cu^+ will fall off from the original capturing site due to the degradation of P atoms on the surface. Then these Cu^+ ions can trigger the Fenton-like reaction. On the other hand, a small fraction of Cu^{2+} will be released from the BPNS to plasma, then react with GSH to generate Cu^+ . This serves as another resource of Cu^+ in the TME to trigger a Fenton-like reaction.

In our experiment, we first demonstrated the redox reaction between Cu^{2+} and BPNS in vitro. Two assays were used to confirm the occurrence of redox reaction and the generation of Cu^+ . One method is XPS and another method is EPR. Both of the experiments unambiguously proved the generation of Cu^+ when exposing BPNS with Cu^{2+} for some time. For cell-based experiments, we didn't probe the generation of Cu^+ in situ in TME due to the shortage of methods to monitor this process. Despite that, we measured the ROS level in cells (Supplementary Fig. 20a). Moreover, as cell apoptosis is a direct result caused by ROS generation, we also performed multiple assays to investigate the cell apoptosis, such as TMRE assay, caspase-3/7 measurement, Apo-BrdU staining. All these results together suggested that the BP@Cu induced higher intracellular ROS level and

results in serious cell apoptosis than BPNS. Therefore, we believe that the redox reaction occurs in TME and the Cu^+ -catalyzing Fenton reaction helps kill the cancer cells.

The description of DCFH-DA staining in manuscript was shown below:

The intracellular ROS generation was further confirmed by DCFH-DA staining (Supplementary Fig. 20a). The brightest fluorescence was observed in $\text{BP@Cu}_{0.4}\text{@PEG-RGD}$ treated cells, indicating the highest ROS concentration in cells. Compared to BPNS treated cells, the $\text{BP@Cu}_{0.4}$ treated cells showed brighter fluorescence, suggesting that the Cu^{2+} plays a vital role in inducing the production of ROS in cells.

Supplementary Fig. 20 (a) Intracellular ROS generation measurement. Cells were treated with PBS, BPNS (100 ppm), $\text{BP@Cu}_{0.4}$ (100 ppm of BPNS), or $\text{BP@Cu}_{0.4}\text{@PEG-RGD}$ (100 ppm of BPNS) for 24 hours. NIR irradiation (808 nm , 1 w cm^{-2} , 2 min) was applied to all cells after incubation for 4 hours. Cells after treatment were loaded with $10\text{ }\mu\text{M}$ DCFH-DA assay for 20 min and then imaged by fluorescence microscopy. Scale bars, $100\text{ }\mu\text{m}$ for all panels.

• *TEM and HRTEM images of BP@Cu and BP@Cu@PEG-RGD will give direct and accurate evidence of the structural integrity of the nanosheet structure.*

Response: Thank you for this comment. As per the reviewer's suggestion, we have performed TEM and HRTEM characterizations of BP@Cu and BP@Cu@PEG-RGD . These data have been added to the supplementary material. Also, we have added some description of these data in the manuscript text accordingly.

The corresponding description in the manuscript text:

Page 5, last paragraph,

... The TEM and HR-TEM imaging studies of BP@Cu were also performed. Many holes appeared on the surface of BPNS (Supplementary Fig. 5a), leading to the blur of the crystal lattice of BPNS in the HR-TEM (Supplementary Fig. 5b). This phenomenon may be caused by the oxidation effect of Cu^{2+} .

Page 6, first paragraph,

The three-dimensional AFM image also confirms the uniform distribution and sheet morphology of BP@Cu@PEG-RGD (Supplementary Fig. 6b). The TEM image of BP@Cu@PEG-RGD shows consistent outcomes with the AFM results (Supplementary Fig. 5c). No serious surface degradation was observed compared to BP@Cu (Supplementary Fig. 5d), indicating the considerable stability of the BP@Cu@PEG-RGD in a normal atmosphere.

New data in supplementary material:

Supplementary Fig. 5 (a) TEM images of BP@Cu_{0.4}. (b) HR-TEM of BP@Cu_{0.4}. The yellow circle indicates the surface oxidation of BPNS. (c) TEM images of BP@Cu_{0.4}@PEG-RGD. (d) HR-TEM of BP@Cu_{0.4}@PEG-RGD. The inset in d shows the SAED pattern.

• According to the design and Fig-1 the material contains both Cu⁺² and Cu⁺¹. However Cu⁺¹ is not a stable agent in the biological environment. Specific coordination complexes, such as TBTA, has ben used to stabilize the +1 oxidation state of the Cu in water. Is there any proof of the Cu⁺¹ state in the material. EPR spectroscopy could be an experimental tool to check the oxidation state.

Response: Thank you for this important comment. In our design, we proposed that the Cu²⁺ ion oxidizes black phosphorus to P_xO_y and itself is reduced to Cu⁺. We provided the XPS characterization data in the original manuscript which demonstrates the generation of Cu⁺ when exposing BP@Cu²⁺ in the atmosphere. These data were shown in Figure 3e. To further confirm the existence of Cu⁺, we performed an EPR experiment based on the reviewer's suggestion. From the EPR result, we can see an intense signal from CuSO₄ because of the paramagnetic property of divalent copper ions, however, BPNS shows no EPR signal (Supplementary Fig. 12a). When exposing the BP@Cu²⁺ in the atmosphere and measuring the EPR spectra at different time points, an apparent decrease of the signal intensity was observed (Supplemental Fig. 12b). This decrease can be attributed to the reduction of Cu²⁺ to non-paramagnetic Cu⁺ charge state. In the revised manuscript, we have added these data in the SI and added some description in the manuscript text accordingly.

The description in the manuscript text :

To further confirm the reduction of Cu²⁺, we performed electron paramagnetic resonance (EPR) spectroscopy. An intense signal appears in the spectrum of CuSO₄ because of the paramagnetic property of divalent copper ions. BPNS shows no signal in the EPR spectrum (Supplementary Fig. 12a). When exposing the BP@Cu²⁺ in atmosphere and measuring the EPR spectra at different time points, apparent decreases of the signal intensities were observed (Supplemental Fig. 12b). These decreases may be attributed to the reduction of Cu²⁺ to non-paramagnetic Cu⁺ charge state.

The data in the SI:

Supplementary Fig. 12 (a) EPR spectra of CuSO_4 and BPNS. (b) EPR spectra of $\text{BP@Cu}_{0.4}$ obtained at various time points to monitor the redox reaction between BPNS and Cu^{2+} at the atmosphere. The decrease in the intensity of Cu^{2+} indicates the reduction of Cu^{2+} to Cu^+ .

• Cyclic peptide *c(RGDyC)* is conjugated to the PEG molecules by thiol-maleimide chemistry. However in general term the click reaction refers the reaction between alkyne and azide. Here it is better to call as thiol-maleimide conjugation than Click reaction.

Response: Thanks for this suggestion. We agree with your comment and changed the “thiol-maleimide click reaction” to the “thiol-maleimide conjugation reaction” in the manuscript.

Original:

The cyclic peptide *c(RGDyC)* is an excellent ligand for $\alpha_v\beta_3$, which was conjugated to the PEG molecules via the straightforward thiol-maleimide click reaction⁶⁸.

Revised:

The cyclic peptide *c(RGDyC)* is an excellent ligand for $\alpha_v\beta_3$, which was conjugated to the PEG molecules via the straightforward thiol-maleimide conjugation reaction⁶⁸.

• Fig 3a: The absorption spectra of BP@Cu_x display an interesting outcome. Author should make a clear interpretation of the fig 3a. The $\text{BP@Cu}_{0.1}$ shows a peak at 800nm, which could be the indication of the better PTA upon irradiation of NIR light. Why it has been recorded after 4h of storage in water? Its not clear what is the aim of the experiment, does it correspond to the photochemical property of the material or degradation of that? Absorption spectra at different time points to correlate the coordination of Cu to BP and degradation will ban interesting topic to discuss.

Response: Thanks for the valuable comments. We apologize for the poor integration of the UV-Vis spectra. In the original manuscript, Fig. 3a was used to explain the faster degradation of BP@Cu_x species than BPNS. Thus, we stored the BP@Cu_x samples in water for 4 hours before absorption measurement. From the results, it can be seen that the Cu^{2+} ions facilitate the degradation of BPNS, which is most likely to be caused by the redox reaction between BPNS and Cu^{2+} . To more clearly decipher the photothermal properties and the degradation of BPNS after coordination with Cu^{2+} , we

have conducted additional experiments to record the UV-Vis spectra of BPNS and BP@Cu_x species at different time points. We also measured the absorption spectra of CuSO₄. We found a very strong absorption at λ 800 nm of 0.25 M CuSO₄ (Supplementary Fig. 8a). After mixing BPNS with Cu²⁺ ions and measuring the absorption spectra immediately, we observed an obvious enhancement of the absorbance intensity of BP@Cu_x compared to bare BPNS (Fig 3a and Supplementary Fig. 8b). These results suggest the better photothermal property of BP@Cu_x than BPNS. To study the influence of Cu²⁺ on the degradation rates of BPNS, we measured the absorption spectra of BPNS and BP@Cu_x after 24 hours of storage in water. All BP@Cu_x samples revealed significantly lower absorption than bare BPNS at the wavelength of 300-1100 nm (Supplementary Fig. 8c), suggesting that the Cu²⁺ can facilitate the degradation of BPNS via redox reaction with BPNS. Taken the above result together, we conclude that Cu²⁺ coordination can improve the photothermal property whilst facilitate the degradation of BPNS. These data have now been included as Fig. 3a and Supplementary Fig. 8 in the manuscript and the SI respectively. We have also added further discussion regarding these results in the manuscript.

The revised description in the manuscript:

An ultraviolet-visible-near infrared (UV-Vis-NIR) spectrometer was used to measure the absorbance of the samples; the corresponding absorption spectra are displayed in Fig. 3a and Supplementary Fig. 8. The coordination of Cu²⁺ on the BPNS shows an obvious enhancement of the absorbance intensity of BP@Cu_x compared to bare BPNS (Fig 3a and Supplementary Fig. 8b), which is partially contributed by the strong absorption of CuSO₄ solution at λ 800 nm (Supplementary Fig. 8a). These results suggest the better photothermal property of BP@Cu_x than BPNS. To study the influence of Cu²⁺ on the degradation rate of BPNS, we measured the absorption spectra of BPNS and BP@Cu_x after 24 hours incubation in water. All BP@Cu_x samples revealed significantly lower absorption than bare BPNS at 300-1100 nm (Supplementary Fig. 8c), implying that the Cu²⁺ can facilitate the degradation of BPNS via redox reaction with BPNS.

Figure 3. Influence of Cu²⁺ ions on the stability and photothermal effects of BPNS. (a) UV-vis absorbance spectra of BPNS, BP@Cu_{0.1}, and BP@Cu_{0.2} and BP@Cu_{0.4} with the same amount of BPNS (100 ppm).

Supplementary Fig. 8 (a) The absorbance spectra of BPNS and BP@Cu_x in ddH₂O at the wavelength of 780-830 nm. (b) UV-Vis-NIR absorbance spectrum of 0.25 M CuSO₄ in double distilled water (ddH₂O). (c) The absorbance spectra of BPNS and BP@Cu_x in ddH₂O after 24 hours incubation at room temperature.

• It is clear that the material degrade with time. What is the rate of degradation in presence of blood or plasma?

Response: Thanks for this important question. Because the BPNS will absorb plasma protein to form BPNS-corona complexes (*Nature Communications*, 2019, 9, 2480), it is inaccurate to study the degradation of BP@Cu@PEG-RGD in plasma or blood by conventional methods, such as UV-vis or DLS. Alternatively, we took the advantages of radioisotope ⁶⁴Cu labeled on the BPNS to monitor the degradation rate. When the BP@⁶⁴Cu@PEG-RGD is disassembled by degradation, the ⁶⁴Cu radioisotope will be released from the nanocomplex. Based on this mechanism, we turned to monitor the free ⁶⁴Cu isotope as a marker of the degradation of BP@⁶⁴Cu@PEG-RGD. We employed the radio-thin layer chromatography (radio-TLC) to examine the free ⁶⁴Cu²⁺ ions. The method detail has been included in the “Supplementary methods”. From this result, it can be seen that the BP@⁶⁴Cu@PEG-RGD was gradually degraded with the increase of incubation time. The half-life is estimated to be 35 hours. The experiment outcomes have now been included as Fig. 6a and Supplementary Fig. 22 and 23 and we have added further discussion regarding these results in the manuscript.

The revised description in the manuscript is shown below:

After labeling, the BP@⁶⁴Cu complex was refined by centrifuge and its stability was examined by radio-thin layer chromatography (radio-TLC) in PBS and mouse serum. As shown in Fig. 6a and Supplementary Fig. S21-23, the BP@⁶⁴Cu was very stable in PBS as less than 5% degradation was observed up to 70 hours. In mouse serum, the BP@⁶⁴Cu@PEG-RGD showed gradual degradation with the increase of incubation time. The half-life in serum is estimated to be 35 hours.

Supplementary Fig. 22 (a) Radio-TLC chromatogram of free $^{64}\text{Cu}^{2+}$ ions in 0.2M EDTA. (b-g) Stability data of BP@ ^{64}Cu @PEG-RGD in mouse serum monitored by radio-TLC chromatogram of at various time points. The radioactivity at different peaks was integrated. The fraction of radioactivity from the intact BP@ ^{64}Cu @PEG-RGD marked in green color was shown in each figure.

Supplementary Fig. 23 Thin-layer chromatography (TLC) plates of BP@ ^{64}Cu @PEG-RGD at various time points when incubating in mouse serum. The images were recorded by an autoradiography machine. Free $^{64}\text{Cu}^{2+}$ ions were used as control.

• Fig 3b: It is not clear that the difference in amount of phosphate ion released for each type of BP@Cu is significant in all the timepoints?

Response: Thanks for pointing out this. As per your comment, we have now included the statistical difference data in the revised Figure 3b, as shown below:

Figure 3. (b) Measurement of phosphate anions by a phosphate assay kit. Degradation of BPNSs and BP@Cu_x with the same amounts of BPNS (50 ppm) after storage in water for different periods of time, producing increasing concentrations of phosphate anions in the supernatant. The data show mean \pm s.d. * $p < 0.05$, ** $p < 0.01$, *** $p < 0.001$, **** $p < 0.0001$, analyzed by two-way ANOVA, followed by Bonferroni multiple comparisons post-test. * indicates statistically significant differences compared with the BPNS group.

• Fig 3f-g: A distinguishable difference in the IT maps can be found in case of BP, BP@Cu0.2, and BP@Cu0.2@PEG-RGD. However, in case of Fig 3f both are almost similar. It will be nice to represent the data of Fig 3f with error bar with appropriate statistical test.

Response: We thank for the comment. We have added error bars and statistical significance data in Figure 3f accordingly. The revised Figure 3f was shown below.

Figure 3. (f) Photothermal heating curves recording the temperature variations of BPNS, BP@Cu_{0.2}, and BP@Cu_{0.2}@PEG-RGD with the same amount of BPNS (20 ppm) dispersed in PBS. The data show mean \pm s.d. n = 3, *p<0.01, ***p<0.001, ****p<0.0001, analyzed by one-way ANOVA, followed by Tukey's multiple comparisons post-test. * indicates statistically significant differences compared with BPNS group. # indicate statistically significant differences between BP@Cu_{0.2} and BP@Cu_{0.2}@PEG-RGD.

• Figure 4a-c: Why the incubation time is different for the 3a and 3b-c? Scale bar is absent in Fig 3b. Appropriate statistical analysis should be performed for Fig 4c.

Response: We thank the reviewer for pointing out this. We apologize for making this inconsistency caused by careless typo error. The incubation time for MTT assay is 48 hours in our experiments. We have unified this in the revised version. We have now added a scale bar in Figure 3b and performed statistical analysis for Figure 4e (the original Fig. 4e). The revised figures are shown as follows.

Figure

4. **Cellular uptake studies and** synergistic effects of BPNS and Cu^{2+} ions on killing cancer cells. (a) Cell viabilities measured by MTT assay. B16F10 cells were subjected to various concentrations of BPNS, $\text{BP@Cu}_{0.2}$, $\text{BP@Cu}_{0.2}\text{@PEG-RGD}$, and $\text{BP@Cu}_{0.2}\text{@PEG-RGD + NIR}$. Error bars indicate standard deviation, $n = 3$. (b) Live-cell differential interface contrast (DIC) imaging of B16F10 cells after incubation with BPNS, $\text{BP@Cu}_{0.2}$, $\text{BP@Cu}_{0.2}\text{@PEG-RGD}$, or $\text{BP@Cu}_{0.2}\text{@PEG-RGD + NIR}$ with the same amount of BPNS (100 ppm) for 48 hours, respectively. Scale bars, 100 μm for all panels. (c) Confocal fluorescence images of live B16F10 cells incubating with PBS, BPNS (100 ppm), $\text{BP@Cu}_{0.4}\text{@PEG-FITC}$ (100 ppm BPNS), or $\text{BP@Cu}_{0.4}\text{@PEG(FITC)-RGD}$ (100 ppm BPNS) for 6 hours. Scale bar, 50 μm for all panels. (d) Relative FITC fluorescence in the cells. Mean \pm s.d., $n = 3$. Student's t -test, two-tailed, $***P < 0.001$. (e) Comparison of the 48 h

cytotoxicity of BP@Cu_{0.1}, BP@Cu_{0.2}, BP@Cu_{0.4}, and BP@Cu_{0.4} plus 2 min of NIR laser irradiation against B16F10 cells. Error bars indicate standard deviation, $n = 3$. $*P < 0.05$, $**P < 0.01$, $****P < 0.0001$, analyzed by two-way ANOVA, followed by Tukey's multiple comparisons post-test. (f) Relative cell viabilities of MDA-MB-231, SCC VII, and U87MG cancer cells after incubation with different concentrations of BP@Cu_{0.4}@PEG-RGD. NIR laser irradiation was applied for 2 min after 4 hours of incubation. Error bars indicate standard deviation, $n = 3$. (g) Treatment schedule for *in vivo* cancer therapy by intratumoral injection of nanomaterials. (h) Relative tumor growth curves of the tumor receiving different treatments. Mice were intratumorally injected with 100 μ L saline, BPNS, or BP@Cu_{0.4}. One group of mice injected with BP@Cu_{0.4} received NIR laser irradiation at a density of 1 W cm⁻² for 2 min at 4 hours postinjection. Mean tumor volumes were analyzed by two-way ANOVA. The data represent mean \pm s.d., $n = 5$, $**P < 0.01$, $****P < 0.0001$, analyzed by two-way ANOVA, followed by Tukey's multiple comparisons post-test. (i) Inhibitory rates of B16F10 tumors at day 14 post-treatment. The data represent mean \pm s.d., $n = 5$. The significance of data was analyzed by the Student's *t*-test, two-tailed, $***P < 0.001$, $****P < 0.0001$.

• Fig 4f has no BP@Cu@PEG-RGD but it is present in fig. 4g. What was the actual experiment and how the fig 4g has been generated?

Response: We apologize for this careless mistake. The actual experiment was performed using BP@Cu + NIR. We have revised this error in Figure 4i (original Figure 4g) The corrected figure is shown as follows.

Figure 4. (i) Inhibitory rates of B16F10 tumors at day 14 post-treatment. The data represent mean \pm s.d., $n = 5$. The significance of data was analyzed by the Student's *t*-test, two-tailed, $***P < 0.001$, $****P < 0.0001$.

• No difference between BP@Cu, and BP@Cu@PEG-RGD in case of cell viability (fig 4a,c) but in case of fig 5a the difference is significant. Can author explain this observation?

Response: Thanks for this comment. The experimental conditions between Figures 4a and 5a are different. Figures 4a and 4c showed the cell viability results tested by MTT assay. For the viability of BP@Cu_{0.2} and BP@Cu_{0.2}@PEG-RGD in figure 4a, the experiment was performed without NIR laser irradiation. Only the group BP@Cu_{0.2}@PEG-RGD + NIR received NIR laser irradiation. For figure 5a, it showed the apoptosis results of cells treated with different reagents. In this experiment, all cells received NIR irradiation for 2 min after incubation for 4 hours. In our design, the NIR irradiation exhibits multiple functions. It not only heats the BPNS to exert PTT effect but also controls the degradation of BPNS. When there is no NIR irradiation, the toxicity of BP@Cu_{0.2} and

BP@Cu_{0.2}@PEG-RGD to B16F10 cells are mainly originated from the chemotherapy and chemodynamic therapy effects. However, when there is in presence of NIR irradiation, the PTT dominates the bioeffect. Because the intracellular uptake of BP@Cu_{0.2} and BP@Cu_{0.2}@PEG-RGD varies largely, the apoptotic rate would vary significantly. We are sorry to make this confusion. In the revised manuscript, we have included more experimental conditions in the figure caption.

• *Propidium iodide (PI) is the commonly used fluorochrome for the cell cycle analysis because of the optimal linear DNA-binding capacity i.e. they bind in proportion to the amount of DNA present in the cell. The membrane penetration plays a crucial role in PI staining. However, the membrane integrity is significantly different in live/early apoptotic cell with dead/late apoptotic cell. Hence, when cells are undergoing through apoptotic stress, the higher intensity of PI can also signify the late apoptotic cells. So, apoptosis can also play a major role in the increase of the cell population with higher PI stain in the data. Author can use some other cell cycle analysis dyes (such as, DyeCycle dyes) to check the cell cycle distribution in case of BP@Cu. The other way of overcoming this is to perform fixation and permeabilization to have an easy access inside the nucleus.*

Response: Thanks for this in-depth and valuable comment. The reviewer gave a very insightful comment on using PI as a staining agent for cell cycle analysis. We agree with the viewpoint of the reviewer. Before we performed cell cycle analysis, we have read a few pieces of literature and adopted the most frequently used protocol for our experiment. In our experiment, we used 66% ethanol to fix/permeabilize the cells overnight. Except for ethanol, some researchers may use Triton-X to permeabilize the cell membrane. During the revision, followed by the reviewer's suggestion, we further employed DyeCycle dyes to check the cell cycle distribution. This dye is used to analyze the cell cycle distribution in live-cell, thus prohibiting the bias caused by the membrane integrity. The cells were treated by the same procedure as for PI staining. Then the live cells were stained using the DyeCycle kit and immediately analyzed by BD FACS. The results stained by DyeCycle dyes are very similar to those acquired from PI staining, as the cell population in the G2/M phase is increased in the order of BP@Cu_{0.4}@PEG-RGD > BP@Cu_{0.4} > BPNS > PBS. Moreover, we also observed some differences in the distribution of the cell phase. Specifically, compared to PI staining, the DyeCycle staining delivered an increased S phase cell population but a smaller G2/M phase cell population for all treated groups. This phenomenon may be ascribed to the accessibility of different dyes to enter the cell membrane or errors between independent experimental operations. To let the readers' full access to these data, we have added these cell cycle data from the DyeCycle stained results in SI in the revised manuscript. We also added some descriptions related to the new cell cycle results in the manuscript.

The revised description and figure were shown below:

Since programmed cell death is connected to cell cycle arrest, we analyzed the cell cycle distribution of B16F10 cells treated with different BPNS reagents using PI staining. As shown in Fig. 5e and 5f, exposure of B16F10 to BPNS increased the percentage of G2/M phase cells (14.9% in G2/M) compared with the percentage of PBS-treated cells (8.1%). Moreover, BP@Cu_{0.4} treatment caused a further larger G2/M phase arrest (28.7%). As expected, cells treated with BP@Cu_{0.4}@PEG-RGD exhibited the highest G2/M phase arrest (33% in G2/M), likely due to the active accumulation of BPNS in cells, mediated by RGD-integrin interaction. Notably, the induction of G2/M phase arrest resulted in a large decrease in the number of cells in the G0/G1 phase. Moreover, the changes in cell cycle distribution were further studied by using DyeCycle dye staining in live cells, which results in very similar outcomes as PI staining, with a small deviation of the cell population in each phase (Supplementary Fig. 19). Specifically, we noticed that the DyeCycle dye staining leads to larger S populations and smaller G2/M populations compared to PI staining. Given that different cell states in

each protocol, this deviation may be caused by the accessibility of different dyes to enter the cell membrane or errors between independent experimental operations.

Supplementary Fig. 19 Cells were treated with PBS, BPNS (100 ppm), BP@Cu_{0.4} (100 ppm of BPNS), or BP@Cu_{0.4}@PEG-RGD (100 ppm of BPNS) for 24 hours. NIR irradiation (808 nm, 1 w cm⁻², 2 min) was applied to all cells after incubation for 4 hours. (a) Cell cycle distribution analysis of B16F10 cells stained with Vybrant Dycycle Orange Stains and analyzed by flow cytometry. The data was processed by the Flowjo program. (b) Summary of cell percentage in each cell cycle phase, (1) PBS, (2) BPNS, (3) BP@Cu_{0.4}, (4) BP@Cu_{0.4}@PEG-RGD.

• Author should give additional evidence for apoptosis, such as, monitoring Caspase or other apoptotic proteins.

Response: Thanks for this comment. A series of caspases are involved in cell apoptosis, some of them act as “initiator” roles in apoptosis and the others act as “effector” caspases. Caspase-3 (7) orchestrate the demolition phase of apoptosis that results in the controlled dismantling of a range of key structures within the cell and its subsequent disposal (*Nat Rev Mol Cell Biol*, 2008, 9, 231). Moreover, one of the most noticeable and specific features of apoptosis is the degradation of the DNA into numerous fragments, often down to multiples of 200 base pairs, driven by the activation of caspase-3. These features make caspase-3 (7) an attractive biomarker of apoptosis. Therefore, we have monitored the activation of caspase-3/7 in cells treated with our nanomaterials according to the reviewer’s suggestion. From these results, it can be seen that both BP@Cu_{0.4} and BP@Cu_{0.4}@PEG-RGD treatment activated caspase-3/7 in B16F10 cells, and the BP@Cu_{0.4}@PEG-RGD showed the strongest effect on activating caspase-3/7. These results have now been included in Figure 5c, 5d, and Supplementary Fig. 17. Also, we have added some corresponding description in the manuscript.

The revised manuscript is shown below:

The cell apoptosis is executed through two major executioner caspases, caspase-3 and caspase-7⁷⁴. Therefore, we examined the caspase-3/7 activation in B16F10 cells after treated with our BPNS nanomaterials. For B16F10 cells treated with PBS or BPNS alone, few green fluorescence positive cells (caspase-3/7 activated cells) were observed. In contrast, the treatment with BP@Cu_{0.4} and BP@Cu_{0.4}@PEG-RGD leads to obviously increased population of green fluorescence positive cells (Fig. 5c, 5d, and Supplementary Fig. 17), indicating the strong effects on activation of caspase-3/7.

The added figures in the manuscript and in SI were shown below:

Figure 5. (c) Activation of caspase-3/7 in B16F10 cells after treatment with PBS, BPNS, BP@Cu_{0.4}, or BP@Cu_{0.4}@PEG-RGD for 24 hours. Scale bars, 100 μ m for all panels. **(d)** Quantification of caspase-3/7 positive cells.

Supplementary Fig. 17 Activation of caspase-3/7 in B16F10 cells after treatment with PBS, BPNS, BP@Cu_{0.4}, or BP@Cu_{0.4}@PEG-RGD for 24 hours. Cells were treated with PBS, BPNS (100 ppm), BP@Cu_{0.4} (100 ppm of BPNS), or BP@Cu_{0.4}@PEG-RGD (100 ppm of BPNS) for 48 hours. NIR irradiation (808 nm, 1 w cm⁻², 2 min) was applied to all cells after incubation for 4 hours. Cells after treatment were loaded with 5 μ M CellEvent Caspase-3/7 green detection reagent for 20 min and then imaged by fluorescence microscopy. Scale bars, 100 μ m for all panels.

• As author claims the apoptosis is initiated by the excess production of the ROS, the involvement of mitochondria can give addition value in the study. The change in mitochondrial membrane potential or Cytochrome-c release can be monitored easily.

Response: Thanks for your valuable comment and suggestion. As you commented, mitochondrial dysfunction has been shown to participate in the induction of apoptosis and has even been suggested to be central to the apoptotic pathway. Indeed, the change of the mitochondria transmembrane potential ($\Delta\psi_m$) can lead to the release of apoptogenic factors and loss of oxidative phosphorylation. (*Apoptosis*, 2003, 8, 115). Therefore, following your suggestion, we measured the $\Delta\psi_m$ using the Abcam's TMRE Mitochondria Membrane Potential Assay Kit (ab113852). This kit uses TMRE (tetramethylrhodamine, ethyl ester) to label active mitochondria. If the $\Delta\psi_m$ is normal, the TMRE will emit bright red fluorescence. In contrast, in apoptotic cells, the $\Delta\psi_m$ is changed or depolarized, the fluorescence will be decreased or even disappear. In our study, the assay was performed according to the manufacturer's protocol. From the results, we found that the B16F10 cells treated with PBS showed bright red fluorescence in the cell plasma. In contrast, only part of the cells treated with BPNS maintains red fluorescence, some of the cells showed no fluorescence. Moreover, the cells treated with BP@Cu_{0.4}@PEG-RGD showed the weakest fluorescence, as shown in Supplementary Fig. 16. These results suggest that the BPNS nanomaterials treatment changes the $\Delta\psi_m$ of B16F10 cells and results in apoptosis of cancer cells. In the revised version, we have added this data in the SI, and also added more discussion in the manuscript.

The discussion about the TMRE assay in the manuscript is shown below:

To confirm the apoptosis of B16F10 cells, we further checked the mitochondria transmembrane potential ($\Delta\psi_m$) changes of the cells, as the changes of $\Delta\psi_m$ can lead to the release of apoptogenic factors and loss of oxidative phosphorylation⁷³. TMRE (tetramethylrhodamine, ethyl ester) was used to label active mitochondria and acts as an indicator for $\Delta\psi_m$. For cells treated with PBS, bright red fluorescence was emitted from the cell plasma. In contrast, the cellular fluorescence intensity was sharply decreased in cells treated with BP@Cu_{0.4}@PEG-RGD, suggesting the loss of $\Delta\psi_m$ of the cells. Notably, the BPNS treatment only caused $\Delta\psi_m$ change in a small portion of cells (Supplementary Fig. 16).

The data added in the SI was shown as follows:

Supplementary Fig. 16 (a) Fluorescence microscopy imaging of TMRE-labeled mitochondria in live B16F10 cells treated with PBS, BPNS, or BP@Cu_{0.4}@PEG-RGD for 24 hours. Cells were treated with PBS, BPNS (100 ppm), BP@Cu_{0.4} (100 ppm of BPNS), or BP@Cu_{0.4}@PEG-RGD (100 ppm of BPNS) for 48 hours. NIR irradiation (808 nm, 1 W cm⁻², 2 min) was applied to all cells after incubation for 4 hours. Cells after treatment were stained with 200 nM TMRE for 20 minutes in media, washed briefly with PBS and immediately imaged. The left arrow images show the enlarged view of selected areas (dashed box) in the right arrow. Scale bars, 50 μ m for all panels. (b) Analysis of cells by flow cytometry after TMRE staining. (c) Semi-quantification of membrane potential ($\Delta\psi$) of cells based on the FACS data. Mean \pm s.d., $n = 3$, the Student's t -test, two-tailed, ** $P < 0.01$, *** $P < 0.005$.

• It is not very clear that why the BP@Cu induce apoptosis. To be more specific, why it induce excess production of ROS? Some experimental evidence or explanation should be required in this section.

Response: Thanks for this constructive comment. We apologize for this unclear point. In this study, we demonstrated that BP@Cu is a better PTA than BPNS, as it not only enhances the photothermal performance of BPNS but also accelerates the degradation of BPNS. Our recent study has revealed that the fast degradation of BPNS in cells could elicit the generation of high concentration ROS (Tao W. et al., *Nano Letter*, 2020, in press). Therefore, excess ROS production could result from the fast degradation of BPNS. More importantly, we incorporated Cu²⁺ into BPNS to enable chemodynamic

therapy (CDT), since that Cu(I/II) is an attractive Fenton-like catalyst to generate ROS. Specifically, in our system, the Cu^{2+} loaded on BPNS can react with BPNS via a redox reaction to generate Cu^+ . The generated Cu^+ can react with local hydrogen peroxide (H_2O_2) to yield toxic hydroxyl radicals ($\cdot\text{OH}$) via a Fenton-like reaction, which has a fast reaction rate in the weakly acidic tumor microenvironment (TME), that are responsible for tumor-cell apoptosis. Moreover, a small fraction of Cu^{2+} will be released from the nanocomplex with the degradation of BPNS and react with the excess glutathione (GSH) in TME. This depletion of GSH in TME could further improve the generation rate of $\cdot\text{OH}$ and relieve tumor antioxidant ability. What's more, the Cu^{2+} produced by Fenton-like reaction, in turn, accelerates GSH depletion, further weakening the capacity of tumor cells scavenging ROS by GSH. This feedback loop of $\text{Cu}^{2+}/\text{Cu}^+$ can turn over persistently to feed the Fenton-like reaction and generates continuous toxic ROS to induce cell apoptosis, this reaction cycle can be elucidated as follows:

Supplementary Scheme 1. Elucidation of the BP@Cu enabling CDT in the tumor microenvironment.

Based on the above explanation, we think the Cu(I/II)-enabled Fenton-like reaction majorly attributed to the induction of excess production of ROS in cancer cells. This kind of CDT has been extensively reported in recent literature, *e.g.* (1) *Angew. Chem.* 2016, 128, 1; (2) *J. Am. Chem. Soc.* 2019, 141, 9937; (3) *Adv. Sci.* 2019, 6, 1801986; (4) *Adv. Funct. Mater.* 2019, 1907954; (5) *Adv. Funct. Mater.* 2019, 1904678; (6) *ACS Nano* 2019, 13, 4267; (7) *J. Am. Chem. Soc.* 2019, 141, 849. In our initial manuscript, we demonstrated the production of excess of ROS/superoxide species by FACS analysis. In the revised manuscript, we have now added more explanations in the manuscript as shown below:

Cu(I/II) is an attractive Fenton-like catalyst to generate highly cytotoxic reactive oxygen species, such as the hydroxide radical $\cdot\text{OH}$, which induce cell apoptosis³⁹. Considering the inherent Cu^{2+} -capturing ability of BPNS, we incorporated Cu^{2+} into BPNS to enable CDT. Specifically, in our system, the Cu^{2+} loaded on BPNS can react with BPNS via a redox reaction to generate Cu^+ . Consequently, the Cu^+ may react with the local H_2O_2 via a Fenton-like reaction to generate toxic hydroxyl radicals ($\cdot\text{OH}$). Moreover, a small fraction of Cu^{2+} will be released from the nanocomplex with the degradation of BPNS and react with the excess glutathione (GSH) in TME, further improving the generation rate of $\cdot\text{OH}$ and relieve tumor antioxidant ability. Besides, the Cu^{2+} produced by Fenton-like reaction, in turn, accelerates GSH depletion, further weakening the ROS scavenging ability of GSH in tumor cells. This unique $\text{Cu}^{2+}/\text{Cu}^+$ feedback loop can turn over persistently to feed the Fenton-like reaction and generates continuous toxic ROS to induce cell apoptosis (Supplementary Scheme 1).

• Fig 6a: author should provide detailed experimental condition and normalization method for this data.

Response: Thank you for this comment. Figure 6a displays the stability results of BP@⁶⁴Cu@PEG-RGD in PBS and mouse serum. These data were acquired by using the centrifuge method. During the revision, to more accurately assess the stability, we have employed two methods, centrifuge separation and radio-TLC, to do the experiments. As for stability in saline, these two methods resulted in similar outcomes. In contrast, as for stability in mouse serum, we found that a considerable discrepancy exists between the groups of data acquired by different methods. Specifically, the intact fraction of the nanocomplex generated from centrifuge separation reveals a bit higher than that from the radio-TLC at all testing time points. This discrepancy is most likely to be caused by the nonspecific binding of ⁶⁴Cu to the serum protein, and the high-speed centrifuge brings those free ⁶⁴Cu to the bottom of the centrifuge tubes, leading to the up deviation of the intact fraction when using the following formula to calculate: (total radioactivity – radioactivity in the filtrate)/total radioactivity. However, this problem can be overcome in the radio-TLC assay. Thus, the results from radio-TLC are more accurate than that of centrifuge separation. In this case, we replaced the Figure 6a with the new data obtained from radio-TLC. The detailed experimental condition and normalization method have now been included in SI. The new Figure 6a and the method are also shown below.

Radio-TLC

The stability of BP@⁶⁴Cu@PEG-RGD in mouse serum was examined by radio-TLC, using aluminum foil-backed silica gel matrix strips as the stationary phase. The BP@⁶⁴Cu@PEG-RGD solution (3.7 MBq) was added to 90 μ L of mouse serum (freshly prepared) and then incubated at 37 $^{\circ}$ C for a designated time. A drop of the sample (2 μ L) was taken out and loaded at a designated origin and left to air dry, before eluting with freshly-prepared 50 mM EDTA solution (pH 7.5) as the mobile phase, taking care that the sample origin was not immersed. Free ⁶⁴Cu ion was chelated by EDTA and moved with the solvent front, whereas ⁶⁴Cu-labeled particles remained at the origin. The TLC strips were scanned by the MARITA—the single trace radioactivity thin-layer-chromatography detector. The peaks in the resulted spectra were integrated and the fraction for the intact BP@⁶⁴Cu@PEG-RGD was read out automatically. The TLC strips were also imaged by the digital autoradiography systems (Fujifilm FLA-5100 scanner with Aida Image analysis software, ai4r Le Beaver Real-time Imaging).

Revised Figure 6a was shown below:

Figure 6. (a) Stability of BP@⁶⁴Cu@PEG-RGD in PBS and mouse serum.

• Author should provide some direct evidence of the uptake of the BP@Cu-RGD. The uptake data presented in the manuscript can also come from the accumulation of BP@Cu on extracellular

membrane. A fluorophore conjugated RGD or PEG on the BP@Cu will give a direct evidence of the cellular uptake.

Response: Thank you for this valuable comment. The cellular uptake of our BPNS-based nanocomplex is a precondition for bioactivities, such as apoptosis induction and cell cycle arrest. Multiple bioassays were used to study and confirm the bioactivity of the nanocomplex, which implies the effective nanocomplex uptake. Of course, the most direct and solid evidence of nanocomplex uptake is fluorescence images of the cells. Therefore, based on the reviewer's suggestion, we performed confocal microscopy to check the uptake. Fluorescein (FITC) conjugated PEG was used to modify BP@Cu. For comparison, the BPNS alone, BP@Cu@PEG-FITC were chosen as controls. The experiments were conducted with live cells. The cell nuclei were stained with Hoechst 33258. The cells were imaged at different magnifications. From these data, we can see that the cells treated with PBS and BPNS show little fluorescence in the cells, in contrast, the cells treated with BP@Cu@PEG-FITC and BP@Cu@PEG(FITC)-RGD show bright fluorescence in the cells. The cellular fluorescence treated by BP@Cu@PEG(FITC)-RGD is higher than that by BP@Cu@PEG-FITC, indicating that the RGD modification increases the cellular uptake. What's more, some fluorescent dots and diffused fluorescence were evenly distributed in the cell plasma, suggesting the endocytic uptake and successful escape of the nanocomplex from the endosome. Even in the nuclei, we can see some fluorescence, this may be due to the nanocomplex entrance through the nuclear pores. We also quantified the cellular fluorescence based on the confocal images using the built-in program of the microscope. All of the data have now been included in Figure 3c, 3d, and Supplementary Fig. 14 and 15. Some description of these data has also been added in the manuscript.

The description in the manuscript was shown below:

The cancer cell killing effect of the nanocomplex prerequisites the effective cellular uptake of the nanocomplex. To confirm the successful entrance of the nanocomplex into cancer cells, we conducted confocal fluorescence imaging in live B16F10 cells. Fluorescein isothiocyanate (FITC) conjugated PEG was used to modify BP@Cu_{0.4}, generating BP@Cu_{0.4}@PEG-FITC and BP@Cu_{0.4}@PEG(FITC)-RGD. Live-cell fluorescent images from 6-hour incubation showed that both of the nanocomplexes enter the cancer cells successfully (Fig. 4c, Supplementary Fig. 14 and 15), and BP@Cu_{0.4}@PEG(FITC)-RGD is more likely to be uptaken by B16F10 cells (Figure 4d). Both fluorescent dots and diffused fluorescence are displayed in the cell plasma, indicating the endocytotic uptake of the nanocomplex. Moreover, slight fluorescence appeared in cell nuclei. This phenomenon may be due to the passive penetration through nuclear pores. Taken together, the fluorescence imaging results confirmed the successful uptake of nanocomplex by cancer cells, and RGD modification enhanced the uptake through the integrin-mediated active targeting.

The revised figure 4c and 4d were shown below:

Figure 5. (c) Confocal fluorescence images of live B16F10 cells incubating with PBS, BPNS (100 ppm), BP@Cu_{0.4}@PEG-FITC (100 ppm BPNS), or BP@Cu_{0.4}@PEG(FITC)-RGD (100 ppm BPNS) for 6 hours. Scale bar, 50 μ m for all panels. (d) Relative FITC fluorescence in the cells. Mean \pm s.d., $n = 3$. Student's t -test, two-tailed, *** $P < 0.001$.

Supplementary Fig. 14 Fluorescence images of live B16F10 cell incubating with PBS, BPNS (100 ppm), BP@Cu_{0.4}@PEG-FITC (100 ppm BPNS), or BP@Cu_{0.4}@PEG(-FITC)-RGD (100 ppm BPNS) for 6 hours. Cells were cultured in 35 mm live-cell imaging chamber overnight. Cells were pretreated with Hoechst 33258 (blue) for 10 min for nucleus staining before subjecting to confocal imaging. FITC-labeled PEG was used to modify BPNS. Scale bars, 400 μ m for all panels.

Supplementary Fig. 15 Confocal images of live B16F10 cell incubating with PBS, BPNS (100 ppm), BP@Cu_{0.4}@PEG-FITC (100 ppm BPNS), or BP@Cu_{0.4}@PEG(-FITC)-RGD (100 ppm BPNS) for 6 hours. Cells were cultured in 35 mm live-cell imaging chamber overnight. Cells were pretreated with Hoechst 33258 (blue) for 10 min for nucleus staining before subjecting to confocal imaging. FITC-labeled PEG was used to modify BPNS. Scale bars, 50 μ m for all panels.

• In case of biodistribution assay the nanoparticles were injected through tail vein. However the % ID/g in blood remains low at initial time points and then gradually increases with time. Author should explain this observation.

Response: Thanks for this insightful comment. As you said, we administrated the nanomaterials intravenously. In common cases, the %ID/g in blood would gradually decrease like most other kinds of nanomaterials. However, our result is that the %ID/g of BP@⁶⁴Cu@PEG-RGD in the blood is persistently increased from 1 hour to 42 hours after *i.v.* injection. This phenomenon is most likely to be caused by the following reasons. First, unlike most of the inorganic nanoparticles reported previously, BPNS is biodegradable *in vivo*, as a result, the size of the BPNS will be decreased. This unique property makes the BP@Cu@PEG-RGD reversibly escape from the reticuloendothelial system (RES) after the initial uptake. Specifically, in our study, the average diameter of BP@Cu@PEG-RGD is about 134 nm. When these nanomaterials were *i.v.* injected into the mouse tail vein, they would be rapidly circulated in the whole body. Followed that, these nanomaterials are preferentially trapped by mononuclear phagocyte-rich organs, such as spleen and liver, due to the large size of the nanosheets (> 35 nm). Indeed, both the PET images and *ex vivo* BioD data support this point (Figure 6d, 6h, and 6i). However, because of the persistent oxidation of the BPNSs, these nanomaterials could become “shrunk” and decrease their size. When the diameter is reduced to under 35 nm, these nanomaterials would undergo reversible escape from the macrophagocytes and reback to blood circulation, and finally be cleared by kidney or other excretion organs (*ACS Nano*,

2015, 9, 6655; *Nanomedicine*, 2010, 5, 523). Actually, the persistent increment of %ID/g in kidney confirmed this assumption. This reversible uptake of BP@Cu@PEG-RGD in RES is a major reason for the increase of %ID/g in blood. Second, in our study, the %ID/g is measured based on the radioactivity of ^{64}Cu , rather than direct from BP@ ^{64}Cu @PEG-RGD. Because of the degradation of BPNS, some ^{64}Cu ions would be released from the BPNS and become free ions or chelated by plasma proteins. With the increase of circulation time, the free ^{64}Cu would be increased and lead to the incremental of %ID/g in blood. Moreover, other factors could also lead to this increment, such as the reabsorption by the intestine.

In the initial version of our manuscript, we had added some discussion about the reversible escape of the nanomaterial from the RES, as shown as follows:

In contrast, the nanotracer reached its highest accumulation in spleen, lung, and liver in the first 3 hours and then was gradually excreted from these organs, reflected by a sharp decrease in intensity from 3 hours to 18 hours post injection (Fig. 6h). Simultaneously, an apparent increase in the kidney and tumor occurred. The excretion of the nanotracer by the liver, spleen, and lung might be ascribed to the oxidation of the BPNS. These organs fill with sufficient oxygen to cause the fast oxidation and degradation of BPNS, rendering the BPNS small enough to escape the reticuloendothelial system and undergo reabsorption by the kidney and tumor⁷⁹.

In the revised manuscript, we have added more description of this phenomenon. The revised version is shown below:

In contrast, the nanotracer reached its highest accumulation in spleen, lung, and liver in the first 3 hours and then was gradually excreted from these organs, reflected by a sharp decrease in intensity from 3 hours to 18 hours post-injection (Fig. 6h). Simultaneously, an apparent increase in the kidney and tumor occurred. The excretion of the nanotracer by the liver, spleen, and lung might be ascribed to the oxidation of the BPNS. These organs fill with sufficient oxygen to cause the fast oxidation and degradation of BPNS, rendering the BPNS small enough to escape the reticuloendothelial system and undergo reabsorption by the kidney and tumor⁸². Notably, unlike most conventional inorganic nanoparticles, the %ID/g in the blood of our nanomaterials persistently increased from 1 hour to 42 hours following *i.v.* injection. This phenomenon may be due to the reversible uptake in the mononuclear phagocytes, where the BP@Cu@PEG-RGD is initially uptaken because of the large size (\square 134 nm). With the oxidative degradation of BPNS, the BP@Cu@PEG-RGD would reduce its size ($<$ 35 nm) and eventually escape from the RES-rich organs, such as spleen and liver, and enter the blood circulation.

• Author has mentioned in the introduction section that the traditional PTAs are related with the kidney and liver problem and the current material will overcome of that problem. According to the data the BP@Cus has high accumulation in Kidney and liver. Does it create similar toxicity? Is it possible to give some evidence about the hepatic and renal toxicity? Additionally, nanostructures also accumulate in the lungs and spleen (example Graphenes, CNTs). The authors should look at accumulation and toxicity to these organs.

Response: Thanks for this constructive and valuable comment. BPNS is biocompatible and degradable, leading to its excellent *in vivo* safety. A series of previous studies have systematically investigated the toxicity of BP-based nanomaterials in mouse models. For example, Shao et al. studied the changes in standard hematology, blood biochemistry, and histology of mice treated with biodegradable BPQDs/PLGA nanospheres. All the data suggested that the BPQDs/PLGA exhibited inappreciable toxicity *in vivo*. (*Nat. Commun.*, 2016, 7, 12967). Furthermore, Tao et al. performed blood routine test and H&E staining of the major organs of healthy mice *i.v.* injected with BP-PEG NSs. No observable side effect or toxicity of the PEGylated BP NSs was found even at a higher dose

of 10 mg kg^{-1} in the *in vivo* toxicity studies. (*Adv. Mater.*, 2016, 201603276) These results consistently demonstrated the little side effect and toxicity of BPNS.

In our study, the BP@Cu@PEG-RGD nanocomplex consists of non-toxic components, therefore, the combination of these materials would not cause unexpected toxicity. To test this assumption, we performed hematology analysis, blood biochemical analysis, and histological analysis of major organs of healthy C57/BL6J mice treated by *i.v.* injected with BP@Cu@PEG-RGD (200 μg BPNS). The blood was taken out on day 1, 3, 15, and 30 days postinjection for hematological and blood biochemical analyses. The major organs, including liver, kidney, heart, spleen, and lung, were collected on day 30 postinjection for H&E staining. These data have now been included in Supplementary Fig. 28-31. As seen from those results, no apparent histological abnormalities and lesions were confirmed in all the tested organs, suggesting negligible organ toxicity (Supplementary Fig. 28). The bodyweight of mice was monitored for 1 month following the injection, and no serious bodyweight loss of the mice was observed (Supplementary Fig. 29). Furthermore, blood biochemical analysis of 9 parameters, four of them are markers for the function of the liver, including TP, AST, ALT, and ALB, two are for heart functions, including LDH and CPK, two for kidney functions, including BUN and CREA, and one for gallbladder function-the TBIL, showed no great difference between the BP@Cu@PEG-RGD and the control group (Supplementary Fig. 30), further demonstrating the good biocompatibility of our nanomaterial. Finally, the standard hematology markers including WBC, RBC, HGB, MCV, MCH, MCHC, and PLT were measured. Compared with the saline-treated group, all the BPNS nanomaterials treated groups displayed similar results and no statistical significance between them (Supplementary Fig. 31). These results indicate that no obvious infection and inflammation were elicited by the nanomaterials.

Taken together, we have demonstrated the safety of BP@Cu@PEG-RGD using multiple assays. These results successfully validated our assumption and laid a good foundation for further clinical translation study of these materials. In the revised manuscript, we added a new paragraph to discuss these results.

The description added to the manuscript is shown below:

The good *in vivo* biosafety is one of the prerequisites of nanomedicine for further clinical translation study. To study the toxicity of BP@Cu nanomaterials, we performed hematology analysis, blood biochemical analysis, and histological analysis of major organs of healthy C57/BL6J mice treated by *i.v.* injected with saline, BPNS, BP@Cu_{0.4}@PEG, or BP@Cu_{0.4}@PEG-RGD separately. The blood was taken out on day 1, 3, 15, and 30 days postinjection for hematological and blood biochemical analyses. The major organs, including liver, kidney, heart, spleen, and lung were collected on day 30 postinjection for hematoxylin and eosin (H&E) staining. No apparent histological abnormalities and lesions were observed in all the tested organs, suggesting negligible organ toxicity of the BP@Cu nanomaterials (Supplementary Fig. 28). The bodyweight of mice was monitored for one month following the injection, and no noticeable body weight loss of the mice was observed (Supplementary Fig. 29). Furthermore, blood biochemical analysis of nine parameters was tested, showing no great difference between the BP@Cu_{0.4}@PEG-RGD and the control group (Supplementary Fig. 30), further demonstrating the good biocompatibility of our nanomaterial. Finally, the standard hematology markers were measured. Compared with the saline-treated group, all the BPNS nanomaterials treated groups displayed similar results and no statistical significance among them (Supplementary Fig. 31). These results indicate that no obvious infection and inflammation were elicited by the nanomaterials. Taken together, the BP@Cu_{0.4}@PEG-RGD display satisfiable *in vivo* biocompatibility in the mouse model. This lay a good foundation for further clinical translation study of these materials.

The supplementary data added in the SI is shown below:

Supplementary Fig. 28 Histology evaluation (hematoxylin and eosin-stained images) of the major organs (heart, liver, spleen, lung, and kidney) collected from mice treated with saline, BPNS, or BP@Cu_{0.4}@PEG-RGD at 15 days post-injection. Scale bars, 50 μm for all panels.

Supplementary Fig. 29 Bodyweight of the mice intravenously injected with (1) saline, (2) BPNS, (3) BP@Cu_{0.4}@PEG, and (4) BP@Cu_{0.4}@PEG-RGD at 1, 3, 15, and 30 days post-injection. The administration dose is 200 μg per mouse according to the mass of BPNS. The results show the mean and s.d. ($n = 3$).

a. Day 1

b. Day 3

c. Day 15

d. Day 30

Supplementary Fig. 30 Blood biochemical analysis of the C57/BL6J mice treated with (1) saline, (2) BPNS, (3) BP@Cu_{0.4}@PEG, and (4) BP@Cu_{0.4}@PEG-RGD at 1 (a), 3 (b), 15 (c), and 30 (d) days post-injection. The administration dose is 200 μg per mouse according to the mass of BPNS. The results show the mean and s.d. ($n = 3$) of alanine aminotransferase (ALT), aspartate aminotransferase (AST), total protein (TP), creatine phosphokinase (CPK), total bilirubin (TBIL), blood urea nitrogen (BUN), creatinine (CREA), albumin (ALB), and lactate dehydrogenase (LDH).

Supplementary Fig. 31 Haematological data of the mice intravenously injected with (1) saline, (2) BPNS, (3) BP@Cu_{0.4}@PEG, and (4) BP@Cu_{0.4}@PEG-RGD at 1 (a), 3 (b), 15 (c), and 30 (d) days post-injection. The administration dose is 200 μg per mouse according to the mass of BPNS. The results show the mean and s.d. ($n = 3$). The recording items are white blood cells (WBC), red blood cells (RBC), hemoglobin (HGB), hematocrit (HCT), mean corpuscular volume (MCV), mean corpuscular hemoglobin (MCH), mean corpuscular hemoglobin concentration (MCHC), and platelets (PLT).

Reviewers' Comments:

Reviewer #1:

Remarks to the Author:

The authors have conducted additional studies and have adequately addressed all my initial queries. It should be accepted.

Reviewer #2:

Remarks to the Author:

The authors have satisfactorily addressed all the questions raised by the reviewers. I have no further comments.

April 22, 2020

Reviewers' comments:

REVIEWERS' COMMENTS:

Reviewer #1 (Remarks to the Author):

The authors have conducted additional studies and have adequately addressed all my initial queries. It should be accepted.

Response: Thank you very much for your kind words after revision.

Reviewer #2 (Remarks to the Author):

The authors have satisfactorily addressed all the questions raised by the reviewers. I have no further comments.

Response: Thank you very much for your nice comment.